# Evaluating the land-surface energy partitioning in ERA5

Brecht Martens[1], Dominik L. Schumacher[1], Hendrik Wouters[1,3], Joaquín Muñoz-Sabater[2], Niko E.C. Verhoest[1], and Diego G. Miralles[1]

[1]Laboratory of Hydrology and Water Management – Ghent University, Coupure links 653, 9000 Ghent, Belgium
[2]European Centre for Medium-Range Weather Forecasts (ECMWF), Shinfield Park, Reading, RG2 9AX, United Kingdom
[3]Flemish Institute for Technological Research – Environmental Modelling Unit, Boeretang 200, 2400 Mol, Belgium

**Correspondence:** Brecht Martens (brecht.martens@ugent.be)

**Abstract.**

Climate reanalyses provide a plethora of global atmospheric and surface parameters in a consistent manner over multi-decadal time scales. Hence, they are widely-used in many fields, and an in-depth evaluation of the different variables provided by reanalyses is a necessary means to provide feedback on the quality to their users and the operational centers producing these data sets, and to help guiding their development. Recently, the European Centre for Medium Range Weather Forecast (ECMWF) released the new state-of-the-art climate reanalysis ERA5, following up on its popular predecessor ERA-Interim. Different sets of variables from ERA5 were already evaluated in a handful of studies, but so far, the quality of land-surface energy partitioning has not been assessed yet. Here, we evaluate the surface energy partitioning over land in ERA5, and concentrate on the appraisal of the surface latent heat flux, surface sensible heat flux, and Bowen ratio against different reference data sets and using different modelling tools. Most of our analyses point towards a better quality of surface energy partitioning in ERA5 than in ERA-Interim, which may be attributed to a better representation of land-surface processes in ERA5, and certainly to the better quality of near-surface meteorological variables. One of the key shortcomings of the reanalyses identified in our study is the overestimation of the surface latent heat flux over land, which – although substantially lower than in ERA-Interim – still remains in ERA5. Overall, our results indicate the high quality of the surface turbulent fluxes from ERA5 and the general improvement upon ERA-Interim, thereby endorsing the efforts of ECMWF to improve their climate reanalysis and to provide useful data to many scientific and operational fields.

## 1 Introduction

The partitioning of available energy at the land surface into sensible and latent heat exerts a strong control on atmospheric boundary layer (ABL) dynamics and informs on the coupling strength between land and atmosphere. It translates variations in the state of the land surface (e.g. soil moisture) into changes in the state of the atmosphere (e.g. cloud formation, near-surface air temperature, and the ABL height), both at local and remote locations (Teuling et al., 2017; Miralles et al., 2016;

Guillod et al., 2015; Taylor et al., 2012; Seneviratne et al., 2010). Hence, surface energy partitioning is a crucial process in the occurrence and development of extreme events such as droughts and heatwaves (Miralles et al., 2018, 2014; Teuling et al., 2010; Seneviratne et al., 2006). An accurate representation of the processes involved in this partitioning in land-surface models is thus essential to advance our understanding of past variations in climate, and leverage our abilities to predict future climate and its impacts on our biosphere (Berg and Sheffield, 2018; Dirmeyer et al., 2017).

Climate reanalyses are data sets describing the past and present state of our climate system and are derived using coupled numerical models in which a vast amount of observations is ingested through a state-of-the-art data assimilation system. They typically cover multi-decadal periods and are produced using a constant model set-up and data assimilation framework (often referred to as the Integrated Forecast System, IFS), resulting in consistent data sets describing the recent state of the atmosphere, ocean, and land surface at the global scale. Therefore, reanalyses are widely used to study past climate, to derive long-term records of essential climate variables, to initialise climate or Earth system models, or to force land-surface models offline. The latter may result in higher-resolution specialised land-surface reanalyses (Muñoz Sabater, 2019; Albergel et al., 2018; Balsamo et al., 2015; Reichle et al., 2011). During the last decade, several climate reanalyses have been produced, such as the Modern-Era Retrospective analysis for Research and Applications version 2 (MERRA-2; Gelaro et al. (2017)) from the National Aeronautics and Space Agency (NASA), the Japanese 55-year ReAnalysis (JRA-55; Kobayashi et al. (2015)) from the Japanse Meteorological Agency (JMA), and the ECMWF ReAnalysis Interim (ERA-I; Dee et al. (2011)) from the European Centre for Medium Range Weather Forecast (ECMWF). Recently, ECMWF released ERA5 (Hersbach et al., 2020), a new global climate reanalysis currently spanning the period 1979–present, which serves as the successor of ERA-I. ERA5 is produced using an enhanced modelling and data assimilation framework and it benefits from the assimilation of a significantly higher number of improved observations compared to ERA-I. In addition, the archive will soon cover the period 1950–present and data will become available with a latency of 2 to 4 days. Finally, data are provided at a higher spatial (31 km vs. 80 km) and temporal (hourly vs. 3-hourly) resolution than ERA-I. Note that in case of ERA-I, the 3-hourly resolution can, in fact, only be obtained by combining forecast and analysis steps (Dee et al., 2011).

The number of studies evaluating the quality of different variables from ERA5 is still limited. Yet, results generally point to improvements upon its predecessor and to a better quality than other existing reanalyses for various surface and atmospheric variables (Hersbach et al., 2020). Tetzner and Thomas (2019), for instance, evaluated several meteorological parameters from ERA5 and ERA-I over the southern Antarctic peninsula, and concluded that the better spatio-temporal resolution at which physical processes are resolved in ERA5 positively affects the representation of these variables. These results were confirmed by Wang et al. (2019), who compared the quality of a similar set of near-surface meteorological parameters from ERA5 and ERA-I by means of in situ validation and a modelling exercise where a thermodynamic sea-ice model was forced with reanalysis data over the Arctic ice sheet. Jiang et al. (2019) and Urraca et al. (2018), on the other hand, validated ERA5 radiation components against in situ measurements and compared their quality to other reanalyses, ground-based observations, and satellite data. Although a small positive bias still remains in ERA5 surface irradiance according to the authors – mainly due to errors in the simulation of cloud properties – it is significantly lower than in ERA-I and MERRA-2, especially at inland locations (Urraca et al., 2018). However, in more complex terrain such as mountainous or coastal regions, high-resolution regional-scale

reanalyses such as the COnsortium for Small-scale MOdeling (COSMO) REAnalysis version 6 (COSMO-REA6) from the German weather service, perform better than ERA5 (Urraca et al., 2018). Also surface wind fields have been shown to be accurately represented in ERA5 (Olauson, 2018), mainly as a result of the relatively high spatial resolution at which physical processes are resolved. Other studies have focused on the validation of vertical profiles of atmospheric properties such as humidity and temperature, typically revealing that the representation of these fields is better in ERA5 than in various other data sets, including its predecessor ERA-I (e.g. Brunamonti et al., 2019; Graham et al., 2019; Zhang and Cai, 2019). Indirect evaluations of variables derived from ERA5 have also been performed through different hydrological modelling studies: Albergel et al. (2018), for instance, compared the quality of ERA-I and ERA5 by forcing the Interactions between Soil, Biosphere, and Atmosphere (ISBA) land-surface model with meteorological parameters derived from both reanalyses and comparing the simulated land-surface parameters from ISBA to independent data from satellite observations and in situ measurements. Based on their study, Albergel et al. (2018) concluded that forcing the model with ERA5 surface meteorology yielded consistently better estimates of hydrological states and fluxes. Finally, Tarek et al. (2020) forced two hydrological models for a large number of catchments across the Continental United States (CONUS) to show the improvements of precipitation and near-surface air temperature from ERA5 upon ERA-I.

Despite the importance of an accurate representation of the processes involved in the surface energy partitioning, at present and to the authors best knowledge, no study has directly evaluated the partitioning of energy in ERA5 into the two major surface turbulent fluxes over land (i.e. the surface sensible and latent heat fluxes). As surface energy partitioning acts as a nexus between the land surface and atmosphere, such an analysis might provide useful insights to further improve the modelling of this coupled system, and to advance the quality of future reanalyses. Therefore, the objective of this study is to evaluate the surface turbulent fluxes (and their ratio; i.e. the Bowen ratio) from ERA5 for the period 1983–2018 at different spatio-temporal resolutions. Several experiments are conducted using various observational data sets and modelling tools to evaluate the spatial and temporal variability of the turbulent fluxes at different scales, ranging from point to catchment-scale and sub-daily to yearly scales. The paper is organised as follows: in Sect. 2 we describe the experimental set-up and the data sets used in this study, and provide a brief overview of the key differences between ERA-I and ERA5. In Sect. 3 we describe the results of our experiments and discuss the quality of surface energy partitioning in both reanalyses; concluding remarks are summarised in Sect. 4.

## 2 Data and methods

### 2.1 Reanalyses data

ERA5 is the latest state-of-the-art reanalysis produced at ECMWF (Hersbach et al., 2020), replacing the widely-used ERA-I (Dee et al., 2011). A first segment of the data set, covering the period 2010–2016, was released early 2017, about a decade after the successful release of ERA-I. Compared to ERA-I, which uses IFS cycle 31r1, ERA5 is produced using an improved version of ECMWF's modelling and data assimilation system (IFS cycle 41r2) and ingests information from a substantially larger volume of improved observations, resulting in a high-quality reanalysis of global atmospheric, oceanic, and land-surface fields at hourly time steps, 137 vertical pressure levels, and horizontal resolution of approximately 31 km. Several advancements

upon ERA-I are expected to affect the surface energy partitioning in ERA5 (Hersbach et al., 2020), including (1) a better forcing of solar irradiance, greenhouse gases, and stratospheric sulphate aerosols, which affect the available energy at the surface that strongly drives the turbulent fluxes, (2) a substantially higher spatial resolution, allowing for a more realistic representation of surface-atmosphere interactions in complex terrain such as mountainous or coastal regions, (3) a more advanced land-surface model, namely the Hydrology Tiled ECMWF Scheme for Surface Exchanges over Land (H-TESSEL), which has a demonstrated high skill to simulate surface turbulent heat fluxes in offline experiments (Balsamo et al., 2015; Albergel et al., 2012; Balsamo et al., 2008), (4) improvements in the atmospheric data assimilation component, mainly affecting the atmospheric forcing of the turbulent fluxes, and (5) an evolved land data assimilation system ingesting both snow and soil moisture observations into the land-surface model of the IFS, improving the land-surface control on the turbulent fluxes.

Here, the surface sensible heat flux, surface latent heat flux, and Bowen ratio derived from both ERA5 and ERA-I are evaluated for the period 1983–2018 (i.e. the period for which reference data are available; see Sects. 2.2–2.4) and across the global land surface. Next to the turbulent fluxes and the Bowen ratio, precipitation, 2-meter air temperature, and surface radiation components (from which the surface net radiation is calculated) are processed. These variables are used to disentangle the role of the improved atmospheric forcing versus the more evolved land-surface model in ERA5. All variables are downloaded at their native spatio-temporal resolutions and temporally aggregated to both 3-hourly and daily time intervals.

## 2.2 Eddy-covariance data

In situ eddy-covariance data of the turbulent fluxes (i.e. the land-surface latent flux and land-surface sensible heat flux) are obtained from the FLUXNET 2015 synthesis data set covering the period 1991–2014. The fluxes are processed as in Martens et al. (2017), including (1) masking of rainy intervals at hourly time steps to remove unreliable measurements due to wet sensors, (2) removing gap-filled data records, and (3) aggregating to both 3-hourly and daily temporal resolutions. Note that for the temporal aggregation, 20% of the higher resolution data within the interval are allowed to be missing. Aiming at the calculation of robust validation statistics, only sites with at least 365 daily records (i.e. at least one full year of data) after masking are retained, resulting in a sample of 143 quality-checked eddy-covariance sites (Fig. 1). About 50% of these selected sites have a record length of more than 10 years, with a maximum of 21 years. Note that the same set of towers is used in the sub-daily (i.e. 3-hourly) and daily evaluations of the turbulent fluxes, making the validation metrics between experiments inter-comparable. As shown in Fig. 1, eddy-covariance sites are not uniformly distributed across the global land surface and hydro-climatic regimes are not equally re-presented within the data set. As most sites are located in the CONUS and Europe, warm and humid regions such as the tropics are only poorly covered. Hence, results presented in this paper should be interpreted with the shortcomings of the FLUXNET 2015 data set in mind, as further discussed in Sect. 3.

The daily Bowen ratio at each eddy-covariance site is calculated as the ratio of the land-surface sensible heat flux and the land-surface latent heat flux. The Bowen ratio might be highly unstable when the turbulent fluxes are small compared to the measurement error of the eddy-covariance system, even at the daily temporal resolution. Therefore, outliers in the in situ time series of the Bowen ratio are masked by removing records outside the following window: $[q_{25} - 1.5(q_{75} - q_{25}); q_{75} + 1.5(q_{75} - q_{25})]$, where $q_{75}$ and $q_{25}$ are the 75% and 25% quantiles of the Bowen ratio time series, respectively (Martens et al., 2016).

Finally, next to the turbulent fluxes, measurements of surface net radiation, near-surface air temperature, and precipitation at the eddy-covariance sites are processed as well using a similar approach as for the turbulent fluxes, except for the masking of rainy intervals. As these variables are typically not recorded at each eddy-covariance site, they are only available at 83 sites in total.

## 2.3 Catchment water and energy-balance data

If changes in water storage are neglected, the catchment-scale latent heat flux can be calculated as precipitation minus river discharge; both averaged over a sufficiently long time period (Miralles et al., 2016; Liu et al., 2014; Wang and Dickinson, 2012; Miralles et al., 2011; Vinukollu et al., 2011). By taking into account the latent heat of vaporisation and the density of water:

$$\lambda \rho E = \lambda \rho (P - Q), \tag{1}$$

where $\lambda$ is the latent heat of vaporisation of water (assumed to be constant; $2260 \cdot 10^3$ J kg$^{-1}$), $\rho$ is the density of liquid water (assumed to be constant; 1000 kg m$^{-3}$), $E$ is terrestrial evaporation (m s$^{-1}$), $\lambda \rho E$ is the land-surface latent heat flux (W m$^{-2}$), $P$ is precipitation rate (m s$^{-1}$), and $Q$ is the river discharge (m s$^{-1}$). The assumption that changes in catchment water storage can be ignored requires the consideration of a sufficiently long period compared to the concentration time of the catchment; often, a yearly aggregation period is considered to be sufficient (see e.g. Miralles et al., 2016).

A similar reasoning as for the catchment mass balance can be made in terms of energy balance: when changes in energy storage can be neglected, the energy balance at the catchment implies that the land-surface sensible heat flux can be calculated as the difference between surface net radiation and the sum of ground and latent heat fluxes:

$$H = R_{\mathrm{n}} - (G + \lambda \rho E), \tag{2}$$

where $H$ is the surface sensible heat flux (W m$^{-2}$), $R_{\mathrm{n}}$ is the surface net radiation (W m$^{-2}$), and $G$ is the ground heat flux (W m$^{-2}$). Combining Eqs. 1 and 2 thus provides a means to evaluate the long-term average catchment-scale Bowen ratio, derived from surface net radiation, ground heat flux, precipitation, and river discharge as:

$$\beta = \frac{(R_{\mathrm{n}} - G)}{\lambda \rho (P - Q)} - 1, \tag{3}$$

where $\beta$ (–) is the Bowen ratio.

In this study, Eqs 1–3 are used in combination with an observational data set of river discharge covering the period 1983–2014 to derive an annual benchmarking data set of turbulent fluxes and Bowen ratio at the catchment scale.

### 2.3.1 Discharge

Discharge measurements are obtained from the Global Runoff Data Centre (GRDC), providing data for nearly 4000 catchments with a daily or monthly temporal resolution. As in Miralles et al. (2016), records with data artifacts are first removed based

on an exhaustive visual screening, and only catchments with an area larger than 2500 km$^2$ are considered. In addition, only catchments with a gridded area (on a regular 0.25° latitude–longitude grid) deviating less than 20% from the area reported by GRDC are retained. If measurements are recorded at multiple locations and thus for different drainage areas (particularly in Central Europe), measurements further downstream are favoured. By doing so, catchments are selected without any spatial

overlap (due to possible sub-catchments measured upstream). After this initial filtering, data available at the daily scale are first aggregated to monthly values, given that at least 25 days per month are present. To reduce the impact of e.g. human disturbances such as large-scale groundwater pumping or regulations of river flow, non-overlapping, centered moving averages containing monthly data of 15 years are calculated as described in Dehghani et al. (2019). Any catchment for which the average of a window is exceeded more than three standard deviations by the mean of the subsequent window are discarded to remove

catchments where obvious disturbances occur during the study period. Finally, monthly averages are aggregated to annual averages, conditioning on at least 10 months per year being present.

### 2.3.2   Atmospheric forcing

Surface net radiation and precipitation to derive catchment-scale validation data for the turbulent fluxes and the Bowen ratio using Eqs. 1–3 are taken from the respective reanalysis in order to mainly evaluate the effect of the land-surface scheme in the

IFS on the surface energy partitioning, rather than the combined effect of the atmospheric and land-surface model. Therefore, the reanalyses data (Section 2.1) are temporally aggregated to the annual resolution and spatially aggregated to the scale of the catchments.

### 2.3.3   Ground heat flux

The ground heat flux is calculated as a fixed fraction of the surface net radiation depending on the land cover as in Martens et al.

(2017, 2016) and Miralles et al. (2011). The land cover is parameterised by the Global Vegetation Continuous Fields product (MOD44B v6; Dimiceli et al. (2015)) derived from measurements of the MODerate-resolution Imaging Spectroradiometer (MODIS). Hence, each grid cell is covered by a certain fraction of tall vegetation (e.g. forests), low vegetation (e.g. grasslands), and bare soil. For the fraction of tall vegetation, the ground heat flux is assumed to be 10% of the net radiation, while for the fractions of low vegetation and bare soil the corresponding percentages are 20% and 35% (Miralles et al., 2011; Santanello and

Friedl, 2003; Kustas and Daughtry, 1990). Altogether, the fraction of net radiation assumed to be converted into the ground heat flux is the weighted average of the former percentages with the fractional land covers.

### 2.4   Balloon soundings

The Integrated Global Radiosonde Archive (IGRA; Durre et al. (2006)) is a data set of direct atmospheric sounding observations from balloons across the globe, representative of different environmental and climate conditions (Wouters et al., 2019) and can

be used to evaluate estimated profiles of atmospheric properties. The data set will be used here to evaluate atmosheric profiles derived from forcing an ABL model (Section 2.6) with ERA5 data. The balloon soundings are screened for the observation

time and quality as in Wouters et al. (2019). A detailed description of this data set, together with a description of the processing and quality checks can be found in Wouters et al. (2019). The data set used in this study consists of approximately 18000 quality-checked morning–afternoon sounding pairs from 121 locations across the globe from 1981 to 2018.

## 2.5 GLEAM

The Global Land Evaporation Amsterdam Model (GLEAM) is a process-based semi-empirical model designed to estimate terrestrial evaporation and its separate components at the global scale from satellite observations alone (Miralles et al., 2011). In summary, GLEAM first calculates potential evaporation using the Priestley and Taylor equation (Priestley and Taylor, 1972) for four land cover fractions per grid cell: (1) low vegetation, (2) tall vegetation, (3) bare soil, and (4) open water. Estimates of potential transpiration (for the first two fractions) are converted into actual transpiration by applying an empirical multiplicative

stress factor. The latter is calculated as a function of vegetation optical depth – which is used as a proxy for vegetation water content (Liu et al., 2013, 2011) – and root-zone soil moisture. The root-zone soil moisture in GLEAM is calculated using a multi-layer soil water balance model driven by precipitation, and is further optimised using a Newtonian Nudging data assimilation scheme (Martens et al., 2017, 2016). For the bare soil fraction, the evaporative stress factor is calculated based on surface soil moisture alone, while for the open-water fraction, no evaporative stress is considered (i.e. actual equals potential

evaporation). Finally, for grid cells covered by snow, sublimation is calculated using the Priestley and Taylor equation with a specific set of parameters (Murphy and Koop, 2005). The fraction of precipitation intercepted by the vegetated surface and directly evaporated back into the atmosphere (i.e. rainfall interception loss) is only calculated for the fraction of tall vegetation. For this purpose, the implementation of Gash's analytical model of rainfall interception (Gash, 1979) by Valente et al. (1997) is used. Ultimately, the total evaporative flux is calculated by summing the fluxes calculated for the four cover fractions. For a

detailed description of GLEAM, we refer the readers to Martens et al. (2017, 2016) and Miralles et al. (2011, 2010).

Here, GLEAM is used as a tool to assess quality differences in some key meteorological drivers of the turbulent fluxes, derived from ERA5 and ERA-I, and to explore the skill of the land-surface model implemented in ERA5 (H-TESSEL) to accurately model the control of the land surface on the turbulent heat fluxes. To do so, GLEAM is forced by an up-to-date version of the GLEAM v3a forcing data base described in Martens et al. (2017), which uses near-surface air temperature and

surface net radiation from ERA-I (hereafter referred to as GLEAM+ERA-I). Next, GLEAM is also forced using the same data set, but with near-surface air temperature and surface net radiation from ERA5 (hereafter referred to as GLEAM+ERA5). Although GLEAM has been designed to target the accurate estimation of terrestrial evaporation (or surface latent heat flux), we also calculate the estimated surface sensible heat flux as the residual of the energy balance, ignoring changes in energy storage (Eq. 2). Based on the estimates of both turbulent fluxes, the Bowen ratio from GLEAM is also calculated. The model is run

for the period 1989–2015 – where 1989 is used as a spin-up year (Martens et al., 2017) – at daily temporal resolution, and on a regular 0.25° latitude–longitude grid (Martens et al., 2017). All inputs, either sourced from ERA-I or ERA5, are processed as in Martens et al. (2017), including a linear re-sampling in both time and space to the spatio-temporal resolution used by GLEAM.

## 2.6 CLASS4GL

The Chemistry Land-surface Atmosphere Soil Slab (CLASS) model for GLobal studies (CLASS4GL; http://class4gl.eu) is a free software tool designed to investigate the dynamics of the ABL and its sensitivity to different land and atmospheric conditions using data from weather balloons (Wouters et al., 2019). The core of CLASS4GL is the ABL model CLASS, which is coupled to a soil-vegetation module allowing the simulation of the diurnal evolution of the ABL with a temporal resolution of 60 seconds. The platform is able to mine appropriate observational data from global radio soundings, satellite data, and reanalysis data from the last 40 years to constrain and initialise the ABL model. Its interactive interface automatises multiple simulations of the ABL in parallel and allows to perform global perturbation experiments. It aims to foster a better understanding of land–atmosphere feedbacks and to disentangle the drivers of (extreme) weather conditions globally.

Here, CLASS4GL is used as a tool to assess whether the surface energy partitioning in ERA5 has been improved upon ERA-I in a similar experiment as described in Sect. 2.5 with GLEAM. Therefore, CLASS4GL is forced with the turbulent fluxes derived from both ERA5 and ERA-I to simulate diurnal tendencies of potential temperature, humidity, and mixed-layer height. As described by Wouters et al. (2019), the evaporative fraction derived from reanalysis data (either ERA-I or ERA5) is used to guide the simulations of the ABL diurnal evolution, and the resulting afternoon profiles of humidity, potential temperature, and ABL height.

## 2.7 Evaluation strategy

### 2.7.1 Evaluation using eddy-covariance data and balloon soundings

Both the turbulent fluxes (and Bowen ratio) from the reanalyses (Sect. 2.1) and the estimates from the GLEAM experiments (Sect. 2.5) are directly compared against the in situ eddy-covariance measurements (Sect. 2.2). For each eddy-covariance site in the validation data set, the variables from the overlapping model grid cells are extracted at their native spatial resolution and both as 3-hourly and daily (temporal) aggregates. Note that for the experiments involving GLEAM, only daily estimates are available. Eddy-covariance sites located within the same model grid cell are treated separately in the validation to avoid potential problems resulting from merging sensors with different absolute values and gaps in their record (Martens et al., 2017). Also note that there is a substantial mismatch between the footprint of the eddy-covariance system and the model grid cells, resulting in a representativeness error that can be a substantial fraction of the total error (Jiménez et al., 2018).

As the temporal variability of the turbulent fluxes is strongly influenced by the seasonal cycle of its main drivers at the scales considered in this experiment, the performance of the land-surface schemes in response to anomalous weather conditions (i.e. with respect to the seasonal cycle) might be masked when raw time series are analysed. As such, the evaluation of the turbulent fluxes against the FLUXNET data set will be done based on standardised anomalies to better evaluate the skill of the reanalyses in capturing the effect of specific meteorological conditions on the surface energy partitioning. Therefore, standardised anomalies of the turbulent fluxes are calculated (and Bowen ratio) from (1) the reanalyses, (2) the GLEAM experiments, and (3) the eddy-covariance measurements prior to calculating validation metrics. Note that the calculation of

standardised anomalies allows to directly compare the quality of the turbulent fluxes and the Bowen ratio, despite their different orders of magnitude.

Anomaly time series are calculated by (1) subtracting for each time interval the expected value (i.e. the climatology), calculated as the multi-annual average for that time interval, and (2) dividing by the standard deviation of the expectation. To calculate climatologies of the eddy-covariance data, only FLUXNET sites with a minimum record length of five years are considered, resulting in 77 eddy-covariance towers for the evaluation of the anomaly time series (Fig. 1).

Using the standardised anomalies of the in situ eddy-covariance measurements as a reference, the Pearson correlation coefficient ($R$) and Mean Absolute Difference (MAD) of the reanalyses data sets and the estimates from GLEAM are calculated to evaluate their quality (Sect. 3.1.1). In addition, the Mean Difference (MD) of the raw data series is calculated to assess the bias in the estimates. Metrics are visualised in violin plots constructed using a kernel density estimation approach with a band width calculated according to Scott (1979). For the MD and $R$, a 95% confidence interval is calculated at each FLUXNET site following the procedure outlined in De Lannoy and Reichle (2016). First, the temporal auto-correlation in both the reference and estimated time series is calculated to correct the degrees of freedom (Gruber et al., 2020). Second, a confidence interval is calculated at each FLXNET site assuming a normal distribution for $R$ (after applying a Fisher Z-transformation to the time series) and a Student $t$-distribution for the MD. Metrics are then assumed to be statistically different at the 5% significance level if their confidence intervals do not overlap. Note that we do not calculate confidence intervals for the MAD, as there are no analytical solutions available for this metric and the calculation thus requires a non-parametric approach relying on computationally heavy Monte Carlo simulations (Gruber et al., 2020). Finally, the confidence intervals for the MD and $R$ are averaged across the FLUXNET data set and the average confidence interval is reported.

In a similar manner as for the GLEAM experiment, the simulations of CLASS4GL (Sect. 2.6) are validated against afternoon profiles from balloon soundings sourced from the IGRA data set (Sect. 2.4). However, the skill of CLASS4GL is evaluated based on the Root Mean Squared Error (RMSE) – rather than MAD – $R$, and MD, all calculated on raw time series, and results are visualised in Taylor plots.

### 2.7.2   Evaluation using catchment energy-balance data

Next to the evaluation of the turbulent fluxes from ERA5 against in situ eddy-covariance measurements, an evaluation against catchment-scale water and energy balance data (Sect. 2.3) is also performed. Given the typical bias in eddy-covariance measurements, especially in case of the surface latent heat flux (Beer et al., 2010), an evaluation of the magnitude of the fluxes should be interpreted with care. On the other hand, the catchment-scale energy-balance data is thought to be less biased, especially at the temporal scales considered in this study, and is therefore better suited to evaluate the magnitude of the fluxes (Miralles et al., 2016).

For each catchment in the data set, the turbulent fluxes of the reanalyses (Section 2.1) are temporally aggregated to the annual resolution and spatially aggregated to the scale of the catchments. Next, the MD between the reference data set and the reanalysis is calculated to assess the magnitude of the surface energy partitioning. Results are spatially visualised in global maps and compared against each other by means of scatter plots.

## 3 Results and discussion

### 3.1 Evaluation using eddy-covariance data

#### 3.1.1 Direct comparison to in situ data

Figure 2 shows violin plots of the MD (raw in situ time series as reference), MAD (anomaly in situ time series as reference), and $R$ (anomaly in situ time series as reference) of the turbulent fluxes and the Bowen ratio against in situ eddy-covariance measurements. Average metrics across the FLUXNET data set and their confidence interval are reported in Table 1. Violin plots are presented for the surface latent heat flux (3-hourly and daily resolution), surface sensible heat flux (3-hourly and daily resolution) and Bowen ratio (daily resolution), for ERA5 (green) and ERA-I (yellow), respectively. As shown, statistics are consistently (and statistically significantly) better for ERA5 than for ERA-I, with typically higher $R$ and lower MAD against in situ measurements, even though the bias (MD) remains relatively similar. This indicates that ERA5 is better capturing the temporal dynamics in surface energy partitioning, both at sub-daily and daily temporal resolutions. Especially for the daily-aggregated surface sensible heat flux, a clear improvement can be seen, with the median $R$ of ERA5 across all reference sites approaching the 75% percentile of the ERA-I distribution. Nevertheless, differences are statistically significant in more sites at the sub-daily scale than at daily resolutions: the Pearson correlation coefficient for the surface sensible heat flux from ERA5 is significantly better (at the 5% significance level) at 63% and 38% of the sites at the 3-hourly and daily temporal resolutions, respectively. ERA-I, on the other hand, is only significantly better in approximately 10% of the sites, while in the remainder of sites, differences are not significant. For the surface latent heat flux and Bowen ratio, improvements are less remarkable, but still consistent, as $R$ is significantly better for ERA5 in 59%, 29%, and 39% of the eddy-covariance sites for the surface latent heat flux (3-hourly and daily resolutions) and the Bowen ratio. The opposite is only true in about 8% of the sites. As shown in Fig. 2, both ERA5 and ERA-I tend to overestimate the surface latent heat flux and underestimate the Bowen ratio. Conversely, the average bias in the surface sensible heat flux is close to zero. However, advances in ERA5 have not been able to make a huge difference in these tendencies, as statistics of ERA5 and ERA-I are close to each other and statistically significant in only 1–2 sites. Notably, for both ERA-I and ERA5, validation statistics are generally better for sensible than for latent heat fluxes (see higher median $R$ and lower MAD for sensible heat fluxes, irrespective of data set and temporal aggregation). Albeit the differences in pre-processing techniques and in the sample of eddy-covariance sites, these results are consistent with those by Balsamo et al. (2015) based on a validation of ERA-I only. When the seasonality is not removed (Fig. A1), turbulent fluxes of ERA5 still outperform those from ERA-I, although differences are smaller. In terms of seasonal cycle, the surface sensible heat flux is not necessarily better estimated than the surface latent heat flux; in fact, statistics are generally worse at daily temporal resolution as shown in Fig. A1.

Figure 3 shows the difference between temporal validation statistics calculated at the anomaly time series (i.e. MAD and $R$) of the surface latent heat flux, surface sensible heat flux, and Bowen ratio from ERA5 and ERA-I. Sites are clustered as a function of mean annual precipitation and near-surface air temperature measured at the corresponding eddy-covariance site. Results are consistent with those in Fig. 2, with an overall higher quality (green color) in the sensible and latent heat fluxes

from ERA5. However, it can be argued that there is a tendency of ERA-I to perform better than ERA5 in warm and dry regimes, especially for the latent heat flux and Bowen ratio. These climates are, nonetheless, only sampled by three eddy-covariance towers and thus results may not be generalised. In addition, conclusions based on the performance in certain climate regimes should be interpreted with care, as FLUXNET sites are not uniformly distributed: mild climates are generally over-represented
and most sites are located in Europe and the CONUS, as shown in Fig. 1 and described in Baldocchi et al. (2001).

Presumably, much of the improvement in surface energy partitioning in ERA5 over ERA-I can be attributed to a better representation of land-surface processes in the more advanced H-TESSEL land-surface model and the improved data assimilation system wrapped around the model. Note that both improvements in the atmospheric data assimilation system (by improving the atmospheric drivers of the turbulent fluxes) and the land-surface data assimilation (by improving the land-surface constraint
on the turbulent fluxes) might affect the turbulent fluxes. The better performance of H-TESSEL – in reference to TESSEL, its antecessor used in ERA-I – was already illustrated by Balsamo et al. (2015), who compared the quality of different land-surface variables from ERA-I and ERA-I/Land over the Northern Hemisphere. ERA-I/Land is in essence an offline simulation of H-TESSEL forced with atmospheric data derived from ERA-I. Although quality differences between ERA-I and ERA-I/Land can not only be attributed to the land-surface scheme but also to the different model set-up (i.e. online vs. offline simula-
tion), Balsamo et al. (2015) argued that most of the improvement was due to the land-surface model. As H-TESSEL is now also implemented in ERA5, analogous improvements can thus be expected in ERA5 over ERA-I regarding the simulation of land-surface variables.

Despite the fact that several studies have shown the high performance of H-TESSEL as compared to TESSEL for simulating a variety of land-surface parameters (e.g. Balsamo et al., 2015; Albergel et al., 2012), Balsamo et al. (2015) also showed that
improvements in the turbulent fluxes of ERA-I/Land over ERA-I could not be uniquely linked to the different land-surface scheme. Hence, the better quality of surface energy partitioning in ERA5 is, most likely, not only owed to an improved parameterisation of the land surface, but also a better quality of the atmospheric drivers, simulated by the coupled atmospheric model, which is constrained by a 4D-variational data assimilation of a large number of quality-controlled observations (Hersbach et al., 2020). The better quality of some key meteorological parameters is confirmed by the results presented in Fig. A3,
which shows violin plots of the validation statistics for surface net radiation, 2-meter air temperature, and precipitation at the FLUXNET sites, for 3-hourly and daily temporal resolutions, respectively. Although statistics from ERA5 are better at both temporal resolutions, especially the sub-daily variability of all three variables has been substantially improved over ERA-I, which may largely be the result of a better modelling of cloud properties in ERA5 (Hersbach et al., 2020).

Finally, as described in Sect. 2.1, one of the key improvements in ERA5 upon its predecessor is the higher spatial resolution
at which atmospheric and land processes are resolved. However, Fig. A2 shows that when ERA5 is linearly re-sampled to the spatial resolution of ERA-I, statistics calculated against eddy-covariance measurements only change marginally. Nevertheless, such an analysis only gives a crude idea of the impact of the spatial resolution as (1) due to non-linear processes and feedback mechanisms, a simple re-sampling of the model output does not properly represent the effect of the high-resolution numerical modelling, (2) the effect is expected to be the highest in complex terrain such as mountainous regions, coastal areas, or
highly-heterogeneous landscapes, which are under-represented in the FLUXNET data base, and (3) representativeness errors –

resulting from the relatively small footprint of eddy-covariance towers as compared to model grid cells – remain considerable at the spatial resolution of ERA5.

### 3.1.2 Evaluation using GLEAM

Forcing GLEAM with meteorological data derived from ERA5 and ERA-I provides a convenient and alternative means to evaluate and compare the quality of the reanalyses. Moreover, it allows an evaluation of the usefulness of ERA5 to drive offline models explicitly designed to estimate land-surface fluxes (in case of GLEAM, terrestrial evaporation). Nevertheless, results of such an experiment should be interpreted with care as errors in the forcing might be compensated for by the model. However, parameters in GLEAM are fully based on literature studies (Martens et al., 2017; Miralles et al., 2011) and are not calibrated, the analysis presented in this study is performed over a large number of sites, and the modelling concepts of GLEAM and ERA-I/ERA5 are substantially different. Hence, it is assumed here that errors in neither ERA-I, nor ERA5 are compensated for by GLEAM.

Figure 4a shows violin plots of the MD (raw in situ time series as reference), MAD (anomaly in situ time series as reference), and $R$ (anomaly in situ time series as reference) of the turbulent fluxes and the Bowen ratio derived from GLEAM against in situ eddy-covariance measurements. The average $R$ and MD, together with their confidence interval, are reported in Table 1. Violin plots are shown for both turbulent fluxes and the Bowen ratio at daily temporal resolution; the violin limbs correspond to GLEAM forced with ERA5 (green) and ERA-I (yellow), respectively. Results presented in Fig. 4a show that the estimates of the surface latent heat flux from GLEAM+ERA5 are consistently better than those from GLEAM+ERA-I, especially in terms of $R$ and MAD, while the bias in both is comparable and close to zero on average. While for the MD, GLEAM+ERA5 is only significantly better in a handful of sites, $R$ is significantly better in 22% of the sites for the turbulent fluxes, and in 3% of the sites for the Bowen ratio. However, in the majority of sites (75% for the turbulent fluxes and 91% for the Bowen ratio), differences in $R$ are not statistically significant. These findings support the ones discussed in Sect. 3.1.1, where it was found that some key meteorological drivers of the surface turbulent fluxes are in fact better represented in ERA5 than in ERA-I. On the other hand, with the exception of the bias, statistics for the surface sensible heat flux and Bowen ratio are slightly worse for GLEAM+ERA5 than for GLEAM+ERA-I, but not statistically significant in terms of $R$, as evidenced by the percentages reported above. Nonetheless, when the seasonal cycle is not removed prior to the analysis (Fig. A4a) GLEAM+ERA5 performs consistently (albeit only slightly) better for all variables, suggesting that the seasonality of the meteorological variables used to force GLEAM is better captured in ERA5 than in ERA-I. Despite the fact that the most prominent differences in quality of the surface latent heat flux from GLEAM+ERA5 and GLEAM+ERA-I can be found in mild climates as indicated in Fig. 5a, there is no clear tendency of GLEAM+ERA5 to perform better under specific climatic conditions. The surface sensible heat flux and Bowen ratio from GLEAM+ERA5, on the other hand, tend to degrade in quality (compared to GLEAM+ERA-I) when the climate gets drier and colder. It should be emphasised here again that GLEAM has been specifically designed to estimate the latent heat flux, thus the surface sensible heat flux – calculated here as the residual from the energy balance – has not been subject to equally extensive validations than its latent counterpart, and is prone to be more uncertain.

The turbulent fluxes and Bowen ratio from GLEAM+ERA5 can also be directly compared to ERA5 to provide a crude evaluation of the skill of H-TESSEL as compared to the simpler land-surface scheme in GLEAM. Figure 4b shows that ERA5 is better capturing the temporal dynamics of the anomalies, generally resulting in lower MAD and higher $R$ for all variables. In terms of $R$, ERA5 is performing significantly better (at the 5% significance level) at 27%, 39%, and 27% of the sites for the surface latent heat flux, surface sensible heat flux, and Bowen ratio, respectively. GLEAM+ERA5 is only performing better in 15%, 9%, and 18% of the sites for the same variables, while in the majority of sites, differences are not significant. Only in terms of the bias, ERA5 is overall performing worse than GLEAM+ERA5 (but again, only significant at a very limited number of sites), especially for the surface latent heat flux, which is consistently overestimated in ERA5 for almost all in situ sites (close to 75% of the sites have a positive bias, Fig. 4b). This results in a median MD of 9 W m$^{-2}$ compared to the slight underestimation of -2 W m$^{-2}$ for GLEAM+ERA5 at daily time scales. The positive bias in the surface latent heat flux from ERA5 is very similar to the one from ERA-I, with a median MD of 10 W m$^{-2}$ across all in situ sites at daily resolutions (Fig. 2). The tendency to overestimate the latent heat flux in ERA-I has been previously reported in different studies (Michel et al., 2016; Miralles et al., 2016; Balsamo et al., 2015; Decker et al., 2012), and important changes in the IFS have thus not been able to mitigate this bias in ERA5. Given the interaction between the coupled atmospheric and land-surface model in the reanalysis, the consistent positive bias in the surface latent heat flux is potentially affected by both components of the modelling framework. Although it is hard to identify the exact cause of this bias, it might be induced by the overestimation of the number of wet days typically found in reanalysis data sets (Beck et al., 2019), combined with precipitation rates that are often underestimated (Beck et al., 2019), and vegetation density that might be overestimated (Král, 2011). This presumably results in an overestimation of the interception loss (Král, 2011), an important component of the total latent heat flux in densely-vegetated regions (Martens et al., 2017; Miralles et al., 2010). Note that this hypothesis is partially supported by our analysis: despite the fact that a positive bias can be found virtually everywhere, the strongest biases are typically found in densely-vegetated sites (not shown). We should emphasise here, however, that biases calculated against eddy-covariance measurements have to be interpreted with care, given representativeness errors resulting from the mismatch in spatial footprint between the grid cell and the instrument, and provided that turbulent heat fluxes are thought to be generally underestimated by the eddy-covariance technique, especially in case of the surface latent heat flux (Beer et al., 2010). When the seasonal cycle is not removed prior to the evaluation (Fig. A4b), GLEAM+ERA5 seems to perform equally good or slightly better than ERA5, indicating that GLEAM+ERA5 is marginally better than ERA5 at capturing the seasonal dynamics (Fig. A4b), but worse at capturing the response of surface energy partitioning to short-term anomalies in meteorological conditions (Fig. 4b). Nevertheless, we would like to highlight that ERA5 is a fully-coupled land-atmosphere system permitting a feedback from the land surface towards the atmosphere, while GLEAM is an offline land-surface model forced with atmospheric variables from ERA5. We note that this coupling between the land surface and the atmosphere might have a substantial impact on the quality of the turbulent fluxes (Draper et al., 2018; Balsamo et al., 2015), potentially explaining the differences between GLEAM+ERA5 and ERA5.

Nonetheless, Fig. 5b shows that for the surface latent heat flux, the better performance of ERA5 over GLEAM+ERA5 is mainly due to its better statistics in relatively wet or cold climatic regimes. In drier regimes and, especially warm regions

(mainly located along the west coast of the CONUS and few eddy-covariance sites in Australia; Fig. 1), GLEAM+ERA5 seems to better capture the anomalies of the surface latent heat flux, which might indicate that H-TESSEL has room to improve the response to water stress. For the Bowen ratio, similar conclusions may be drawn, even though the quality of the sensible heat flux in ERA5 is consistently better than in GLEAM+ERA5.

## 3.2  Evaluation using catchment energy-balance data

As described in Sect. 2.3, observations of river discharge may be combined with precipitation, net radiation, and ground heat flux to derive catchment-scale and long-term estimates of the surface turbulent fluxes and the Bowen ratio, providing an alternative means to evaluate the surface energy partitioning in ERA-I and ERA5. Figure 6 compares the percentage MD (%MD, i.e. MD divided by the mean of the reference data set) of the surface latent heat flux, surface sensible heat flux, and Bowen ratio (observations of catchment-scale variables as reference) from ERA5 and ERA-I using a scatter plot. The results shown in Fig. 6 largely correspond to the ones shown in Fig. 2 for the MD and point again to a substantial overestimation of the surface latent heat flux from ERA-I; in 83% of the catchments, a positive bias is obtained. Conversely, the surface sensible heat flux is generally underestimated (a negative bias is found in 61% of the catchments), resulting in an underestimation of the catchment-scale Bowen ratio as well (a negative bias is found in 80% of the catchments). While absolute biases for the surface latent heat flux from ERA5 are lower than from ERA-I (an improvement is found in 75% of the catchments), ERA5 still overestimates the flux in most catchments, as also indicated by Hersbach et al. (2020). More striking are the results for the surface sensible heat flux: while ERA-I generally underestimates the flux, ERA5 overestimates it in about 70% of the catchments. In addition, the absolute bias of the surface sensible heat flux from ERA5 is higher than in ERA-I in 55% of the catchments. However, this potential overestimation is not confirmed by the in situ validation presented in Sect. 3.1.1 (Fig. 2), where the surface sensible heat flux from both reanalyses appeared nearly unbiased. Finally, for the Bowen ratio, estimates of ERA5 appear better in about 60% of the catchments, arguably reflecting the improvement in the surface latent heat flux. Note that a rather strong overestimation of the surface latent heat flux was also found in other reanalyses such as NASA's MERRA and MERRA-2 (Draper et al., 2018). However, in the latter reanalyses, both surface turbulent fluxes were consistently overestimated which could potentially be linked to a positive bias in the incoming radiation at the land surface.

Figures 7–9 show maps of the catchment-scale %MD of the surface latent heat flux (Fig. 7), surface sensible heat flux (Fig. 8), and Bowen ratio (Fig. 9) for ERA5, ERA-I, and the difference in their absolute values. While ERA-I overestimates the surface latent heat flux virtually everywhere, biases are relatively larger in the east of the CONUS and the south of Europe (in regions like Spain and the south of France). In these regions, a strong reduction in bias can be observed for ERA5. Despite the complex interactions between the land surface and the atmosphere in the IFS, these improvements can potentially be related to an improved representation of precipitation in ERA5 as shown by Hersbach et al. (2020) and affecting (1) interception loss in radiation-limited regions such as the east of the CONUS – which might represent a substantial portion of total evaporation in forested regions (Martens et al., 2017; Miralles et al., 2011, 2010) – and (2) the land-surface constraint on terrestrial evaporation in water-limited evaporation regimes like the south of Europe. Note that the latent heat flux in the latter regions will also be strongly affected by improvements in the land-surface data assimilation system (Hersbach et al., 2020; Balsamo et al., 2015).

Over large parts of Europe and western Russia on the other hand, the surface latent heat flux from ERA5 is nearly unbiased, while the overestimation in other regions still remains, albeit reduced compared to ERA-I. Except for a small number of catchments in the northeast of Brazil and the west of the Sahel, the bias of the surface latent heat flux is lower in ERA5 than in ERA-I. The surface sensible heat flux from ERA-I is typically underestimated in high latitudes and the eastern part of the

CONUS, while an overestimation can be seen in most other regions. However, as discussed in the previous paragraph, the bias in the surface sensible heat flux of ERA5 is typically higher, especially over Europe, western Russia, and the east of the CONUS, regions where the bias in the surface latent heat flux is reduced in ERA5. Finally, in absolute terms, the bias in the Bowen ratio increases from ERA5 to ERA-I as evidenced in Fig. 9, and largely follows the patterns set by the bias in the surface sensible heat flux (Fig. 8).

Finally, it should be emphasised here that the quality of the catchment-scale sensible heat flux (and Bowen ratio) estimates used as reference is potentially lower than that of the surface latent heat flux, as (1) the assumption that the ground heat flux is a fixed fraction of the surface net radiation only affects the estimates of the sensible heat (Eq. 2), and (2) the estimates of sensible heat flux depend on the estimates of surface latent heat (Eq. 2), resulting in a propagation of errors which is difficult to assess. Hence, the catchment-scale evaluation of the surface sensible heat flux and Bowen ratio should be more carefully

interpreted.

## 3.3   Evaluation using CLASS4GL

Figure 10 shows the validation of the estimated afternoon ABL properties from CLASS4GL forced with the surface energy partitioning from ERA-I (on the one hand) and ERA5 (on the other hand). The validation is performed by comparison against a global archive of balloon soundings (Sect. 2.6). Results are shown for the diurnal temporal change (tendency) of potential

temperature ($d\theta/dt$), humidity ($dq/dt$), and mixed-layer height ($dh/dt$). The overall performance at reproducing the diurnal ABL tendencies is improved when CLASS4GL is forced with ERA5 instead of ERA-I. This is the case for all statistical scores being considered and for each ABL variable being analysed. In addition, this is also the case in most Köppen–Geiger climate classes, which suggests that the higher performance is consistent across climate regimes. The largest improvement in simulated ABL properties is found for the tendency of specific humidity, where the bias is reduced from 0.10 to 0.05 g kg$^{-1}$ h$^{-1}$ when

CLASS4GL is forced with ERA5 instead of ERA-I. Most of the improvement can be found in days where the mixed layer tends to dry out during the diurnal growth (i.e. negative tendency of specific humidity) and is most likely related to the substantially lower bias in surface latent heat flux from ERA5, as discussed in Sects. 3.1.1 and 3.2. Also, the Pearson correlation coefficient (0.37 vs. 0.50), normalised Root Mean Squared Error (RMSE; 0.22 vs. 0.17 g kg$^{-1}$ h$^{-1}$), and normalised standard deviation (1.2 vs. 1.03) point towards improvements of the ABL simulations when forced by the surface energy partitioning from ERA5.

For the other variables ($dh/dt$ and $d\theta/dt$), improvements are only minor, but still consistent. These results highlight that the surface energy partitioning in ERA5 can lead to improved skill in the diurnal ABL simulations by mixed-layer models such as CLASS.

### 3.4 Global patterns of surface energy partitioning

Figure 11 shows maps of the multi-annual average of the surface latent heat flux, surface sensible heat flux, and Bowen ratio from ERA5 and ERA-I, as well as the difference between both. In both data sets, the expected geographical patterns set by the general climatic conditions emerge. High values for the surface latent heat flux can be found around the equator where both the availability of water and the supply of energy are high, while the lowest values can be found in arid regions such as the Sahara desert, central Australia, the Namibian desert, and the Gobi desert. In terms of surface sensible heat flux, an opposite pattern is shown, with relatively lower values in the tropics, where most of the available energy is consumed to evaporate water, and very high values in the deserts, where virtually no water is evaporated. The Bowen ratio clearly marks the tropical forests and deserts; with intermediate values for mild climates such as central and western Europe.

The globally-averaged surface sensible heat flux from land amounts to 27.2 W m$^{-2}$ and 26.9 W m$^{-2}$ for ERA5 and ERA-I, respectively; a difference of only 1.1% (ERA-I as reference). For the surface latent heat flux, the difference is higher and sums up to -5.2% (ERA-I as reference), with global averages of 44.1 W m$^{-2}$ and 46.5 W m$^{-2}$ for ERA5 and ERA-I, respectively. The latter two values correspond to a yearly total volume of evaporated water of approximately 97.8·10$^3$ km$^3$ and 103.1·10$^3$ km$^3$. Similar values typically found in literature – although based on different land-surface models or retrieval algorithms, input data sets, or region considered (e.g. areas permanently covered by snow or ice included or not) – range between 55·10$^3$ km$^3$ and 80·10$^3$ km$^3$ (Miralles et al., 2016; Wang and Dickinson, 2012, and references therein), pointing towards an overestimation of the total volume of evaporated water in both ERA-I and ERA5. In terms of globally-averaged energy fluxes, the turbulent fluxes from both reanalyses lie within (or close to) the uncertainty ranges reported by Wild et al. (2015), who inferred the magnitude of the global energy fluxes based on a detailed analysis of a variety of observations and model-based estimates. However, the surface sensible heat flux from both reanalyses can be found near the lower boundary of the interval, while the surface latent heat flux may be found near the upper limit of the interval. This is also the case when compared to values reported in Draper et al. (2018) who analysed the turbulent fluxes of NASA's reanalyses products MERRA, MERRA2, and MERRA-Land. They found values for both fluxes ranging between 42 W m$^{-2}$ and 50 W m$^{-2}$, depending on the reanalysis considered. These results confirm our findings in Sects. 3.1 and 3.2 and are in line with results previously reported in literature (e.g. Miralles et al., 2016; Wild et al., 2015; Mueller et al., 2013; Jiménez et al., 2011, and references therein) where similar biases were found for ERA-I.

Figure 11 shows that the lower globally-averaged surface latent heat flux from ERA5 mainly results from reduced values along the east coast of the CONUS, the south of Europe, the Sahel, India, and large parts of South America. These regions align well with the areas identified in Miralles et al. (2016) where ERA-I seemed to strongly overestimate the surface latent heat flux, and thus point to a better performance of ERA5 in these specific regions, although positive biases still prevail (Fig. 7). The surface latent heat flux from ERA5 is higher than the one from ERA-I only in a few areas, such as the central CONUS, eastern Australia, and eastern Europe. For the surface sensible heat flux, differences between ERA5 and ERA-I are clearly defined, with substantially higher values in the equatorial forests and lower values in (semi-)arid regions in the case of ERA5.

## 4 Conclusions

This study evaluated the surface energy partitioning over land in ECMWF's latest reanalysis ERA5 by assessing the quality of the surface latent heat flux, surface sensible heat flux, and Bowen ratio at different spatio-temporal scales and using different validation approaches. Results were also compared with the predecessor ERA-I for reference. Different in situ validation data sets – including eddy-covariance, river discharge, and balloon sounding data – were used to validate the reanalysis fields, and GLEAM and CLASS4GL were adopted as modelling tools to evaluate the surface energy partitioning in both reanalyses.

In a first experiment, the turbulent fluxes and the Bowen ratio from the reanalyses were directly compared against eddy-covariance measurements from the FLUXNET 2015 data set. The analysis revealed that ERA5 performed consistently better than ERA-I for all variables analysed, both at daily and sub-daily temporal resolutions, resulting in lower MAD and higher $R$ against in situ data. The differences were most clear when anomaly time series were analysed, indicating that – although statistics also improved in case of the raw time series – ERA5 is substantially better capturing the response of surface energy partitioning to specific meteorological events. As one of the key changes in ERA5 is the use of the state-of-the-art H-TESSEL land-surface model and improvements in the land-surface data assimilation system, an important part of the improvements may be attributed to the improved land parameterisation. However, a validation of some key meteorological variables against in situ measurements also showed better quality of these parameters from ERA5 than from ERA-I. These results were largely confirmed by an experiment where GLEAM was forced with meteorological fields retrieved from both reanalyses, showing a higher quality of the output based on ERA5 forcing data. Finally, although ERA5 did not seem to perform particularly better than ERA-I in specific climates, it was shown that GLEAM forced with ERA5 meteorology performed better than ERA5 in terms of estimating the surface latent heat flux in warm and dry regimes, indicating possible shortcomings in the land-surface scheme to capture the response of surface energy partitioning to heat and drought stress in ERA5.

In a second experiment, catchment-scale turbulent fluxes derived using discharge, precipitation, net radiation, and ground heat flux data were used to verify the bias in the annual turbulent fluxes from ERA-I and ERA5. Here, a substantial overestimation of the surface latent heat flux from ERA-I became evident. On the other hand, the surface sensible heat flux appeared generally underestimated. While the biases in ERA5 for the surface latent heat flux were found to be lower – a strong reduction was found along the east coast of the CONUS and in the south of Europe – a general tendency to overestimate the latent heat flux still remains in ERA5. In case of the surface sensible heat flux on the other hand, the sign of the bias reversed (i.e. in ERA5 the flux tends to be overestimated) and increased in absolute value.

A better quality of the surface energy partitioning in ERA5 was also confirmed by an experiment where CLASS4GL was forced with the evaporative fraction from ERA-I and ERA5. Simulations of the diurnal evolution of the ABL were validated against a global archive of balloon soundings. CLASS4GL forced with ERA5 showed an overall better skill for simulating the diurnal boundary layer dynamics than when forced with ERA-I. Especially in reproducing the tendencies of specific humidity, CLASS4GL seemed to strongly benefit from the seemingly better surface energy partitioning in ERA5, resulting in a substantially lower bias. The latter could be attributed to the lower bias in the surface latent heat flux in ERA5 than in ERA-I. Since ERA5-forced experiments better explained the global variability of the boundary layer dynamics, this experiment confirmed

the overall better surface energy partitioning in ERA5 than in ERA-I, in line with the other independent experiments presented here.

Finally, the global patterns of turbulent fluxes and Bowen ratio were analysed, and the globally-averaged magnitude of the fluxes was compared with values reported in literature. While the spatial patterns are realistic in both data sets, and align with the expectations from the major hydro-climatological regions, the substantial overestimation of the surface latent heat flux in both reanalyses emerged again. However, the magnitude of the surface latent heat flux was found to be about 5% lower in ERA5 than in ERA-I, pointing towards the reduction of the bias, while the surface sensible heat flux only increased approximately 1%. The main reductions in the surface latent heat flux were found in regions that had previously been highlighted in literature as hotspots of overestimation in ERA-I, such as the south of Europe, the Sahel, India, large parts of South America, and the east coast of the CONUS.

In this paper, a variety of methods and data sets were used to evaluate the quality of the turbulent fluxes (and near-surface meteorology) from ERA5. As discussed throughout the manuscript, all techniques and reference data sets come with their own uncertainties and are derived based on different assumptions leading to potential flaws in the analyses presented in this paper. Eddy-covariance sites in the FLUXNET data set are not uniformly distributed across the globe, neither are the discharge measurements and balloon soundings used in this study. Therefore, conclusions should not be extrapolated to regions that are under-represented in these data sets. In addition, the quality of each reference data set is affected by measurement errors and uncertainties introduced by assumptions made during the processing. Finally, both GLEAM and CLASS4GL are models and cannot be treated as ground truth as their estimates are impacted by uncertainties introduced by the model structure and parameterisation, as well as their inputs. Nevertheless, most analyses point into the direction of improvements from ERA-I to ERA5, irrespective of the validation technique or reference data set used, giving confidence to the conclusions draw in this study. In summary, it can be concluded that – based on the validation data and tools used in this study – the quality of the turbulent fluxes (and near-surface meteorology) from ERA5 has been improved. Although biases (especially in the surface latent heat flux) still prevail, changes in the IFS from ERA-I to ERA5, and improvements in the observational data sets that are assimilated into the models, have thus generally resulted in a higher-quality surface energy partitioning in the reanalysis.

*Code and data availability.* All data sets used in this study can be freely accessed from their respective repositories after registration. ERA-I data were downloaded from the ECMWF web page (https://apps.ecmwf.int/datasets/data/), ERA5 data were retrieved from the Copernicus Climate Data Store (https://cds.climate.copernicus.eu/), GLEAM data were accessed from https://www.gleam.eu/, GRDC discharge data can be downloaded from https://www.bafg.de/GRDC/EN/02_srvcs/21_tmsrs/riverdischarge_node.html, the FLUXNET2015 Tier2 data set can be accessed from the FLUXNET data portal at https://fluxnet.fluxdata.org/data/fluxnet2015-dataset/, input data for CLASS4GL is available at https://www.CLASS4GL.eu/, and the output of CLASS4GL is available upon request. The source code of CLASS4GL can be accessed at https://www.CLASS4GL.eu/.

**Table 1.** Averaged metrics and their confidence interval of surface energy partitioning from ERA5, ERA-I, GLEAM+ERA5, and GLEAM+ERA-I across the FLUXNET 2015 data set.

| | | $\lambda\rho E$ (3h) $\mathrm{W\,m^{-2}}$ | $\lambda\rho E$ (24h) $\mathrm{W\,m^{-2}}$ | $H$ (3h) $\mathrm{W\,m^{-2}}$ | $H$ (24h) $\mathrm{W\,m^{-2}}$ | $\beta$ (24h) $-$ |
|---|---|---|---|---|---|---|
| MD | ERA5 | 9.27 ($\pm$0.080) | 8.49 ($\pm$0.178) | -2.60 ($\pm$0.010) | -2.99 ($\pm$0.140) | -0.56 ($\pm$0.013) |
| | ERA-I | 11.12 ($\pm$0.079) | 10.29 ($\pm$0.180) | -3.38 ($\pm$0.099) | -3.66 ($\pm$0.147) | -0.69 ($\pm$0.012) |
| | GLEAM+ERA5 | n.a. | -3.27 ($\pm$0.176) | n.a. | -5.83 ($\pm$0.153) | -0.25 ($\pm$0.014) |
| | GLEAM+ERA-I | n.a. | -3.76 ($\pm$0.179) | n.a. | -10.14 ($\pm$0.158) | -0.39 ($\pm$0.014) |
| $R$ | ERA5 | 0.34 ($\pm$0.002) | 0.41 ($\pm$0.005) | 0.46 ($\pm$0.002) | 0.50 ($\pm$0.004) | 0.39 ($\pm$0.006) |
| | ERA-I | 0.31 ($\pm$0.002) | 0.39 ($\pm$0.005) | 0.42 ($\pm$0.002) | 0.45 ($\pm$0.004) | 0.36 ($\pm$0.006) |
| | GLEAM+ERA5 | n.a. | 0.35 ($\pm$0.005) | n.a. | 0.45 ($\pm$0.005) | 0.39 ($\pm$0.006) |
| | GLEAM+ERA-I | n.a. | 0.32 ($\pm$0.005) | n.a. | 0.46 ($\pm$0.005) | 0.40 ($\pm$0.007) |

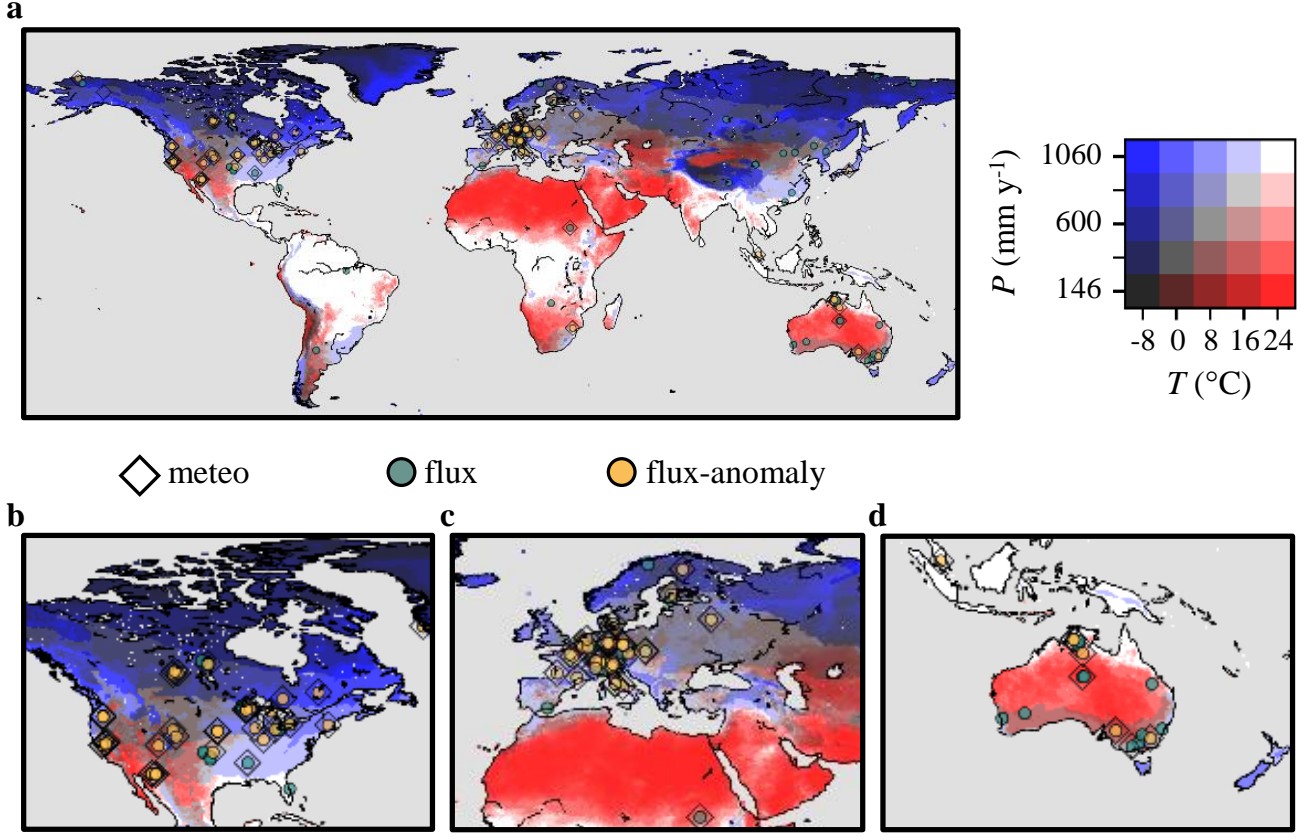

**Figure 1. (a)** Location of the selected eddy-covariance sites. **(b)–(c)** show a detailed view of the sites across the CONUS **(b)**, Europe **(c)**, and Autralia **(d)**. Sites with a record length of less than 5 years (i.e. where no anomalies are calculated) are plotted in green and sites with a record length of more than 5 years (i.e. where anomalies are calculated) are plotted in yellow. Sites where measurements of meteorological data are also available are indicated with a diamond. The background provides information on the climatological mean temperature and precipitation derived from ERA5 (1983–2018).

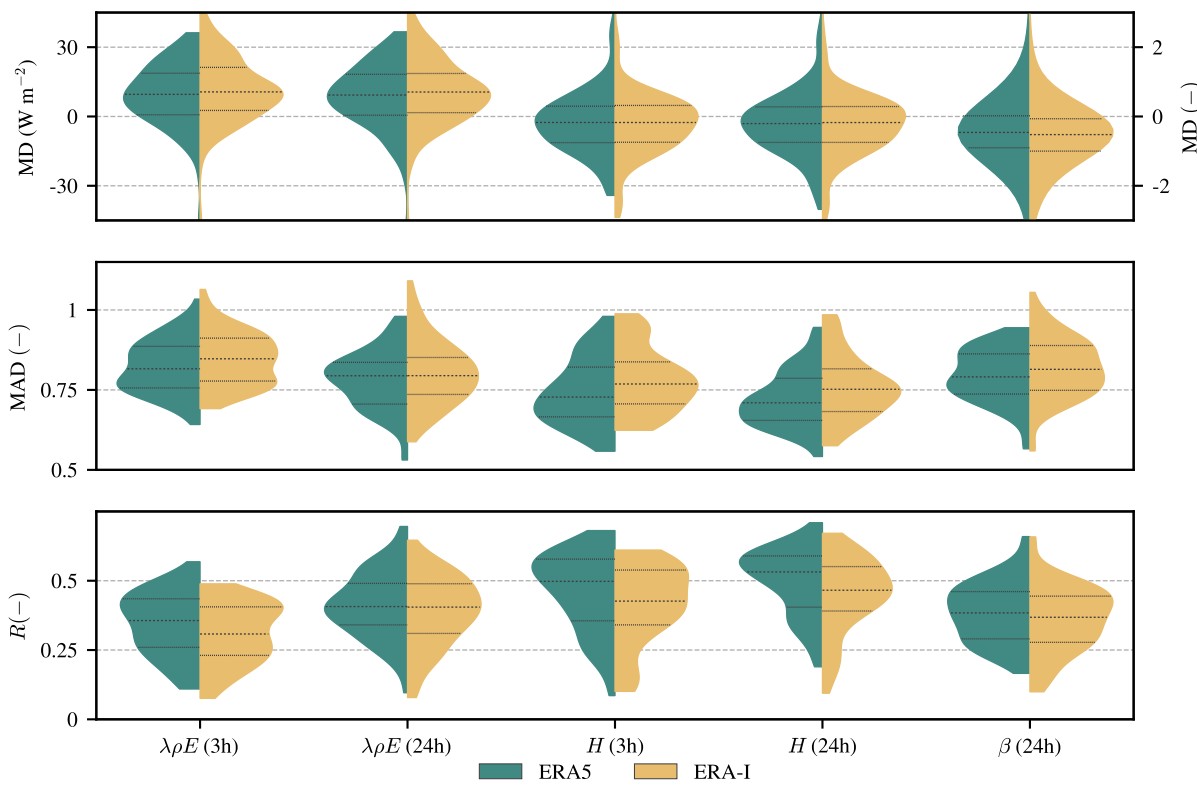

**Figure 2.** Violin plots of temporal validation statistics of the surface latent heat flux ($\lambda\rho E$), surface sensible heat flux ($H$), and Bowen ratio ($\beta$) from ERA5 (green) and ERA-I (yellow). Statistics are calculated against in situ eddy-covariance measurements at both 3-hourly and daily temporal resolutions. Violin plots represent the distribution of the individual validation statistics with indication of the median and inter-quartile range, and are calculated using a kernel density estimation approach. Statistics include the Mean Difference (MD, raw in situ time series from 143 sites as reference), Mean Absolute Difference (MAD, anomaly in situ time series from 77 sites as reference), and the Pearson correlation coefficient ($R$, anomaly in situ time series from 77 sites as reference). The distribution of the MD of $\beta$ is plotted on the right y-axis.

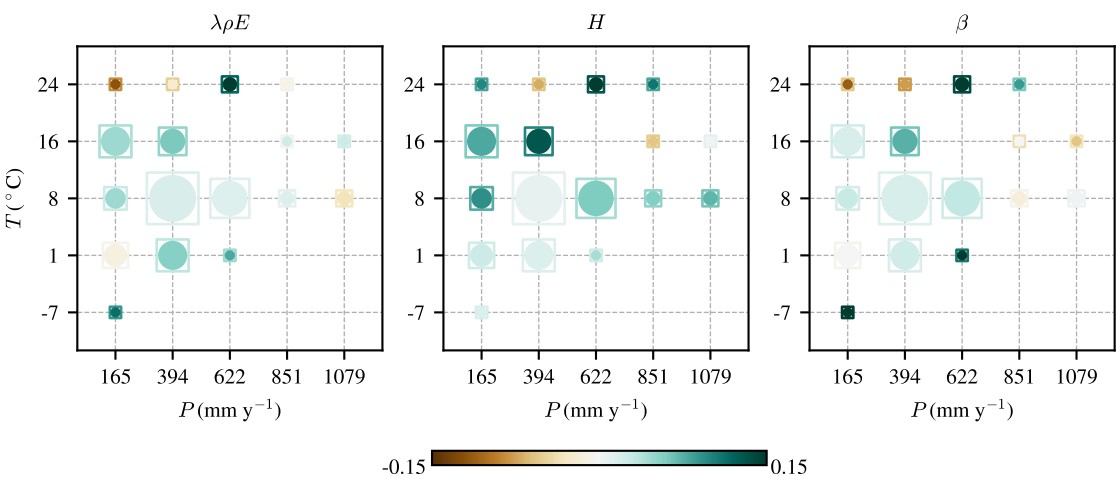

**Figure 3.** Difference between temporal validation statistics of the surface latent heat flux ($\lambda\rho E$), surface sensible heat flux ($H$), and Bowen ratio ($\beta$) from ERA5 and ERA-I grouped as a function of precipitation rate ($P$) and near-surface air temperature ($T$) calculated at the in situ site. Statistics are calculated against in situ eddy-covariance measurements at daily resolution and then averaged across the sites within each group. Statistics include the Mean Absolute Difference (MAD, anomaly in situ time series from 77 sites as reference) and the Pearson correlation coefficient ($R$, anomaly in situ time series from 77 sites as reference). Circles show the $R$ from ERA5 minus the one from ERA-I, while squares show the MAD from ERA-I minus the one from ERA5; hence, green colors represent better statistics for ERA5 compared to ERA-I. The size of the symbols relates to the number of in situ sites per group.

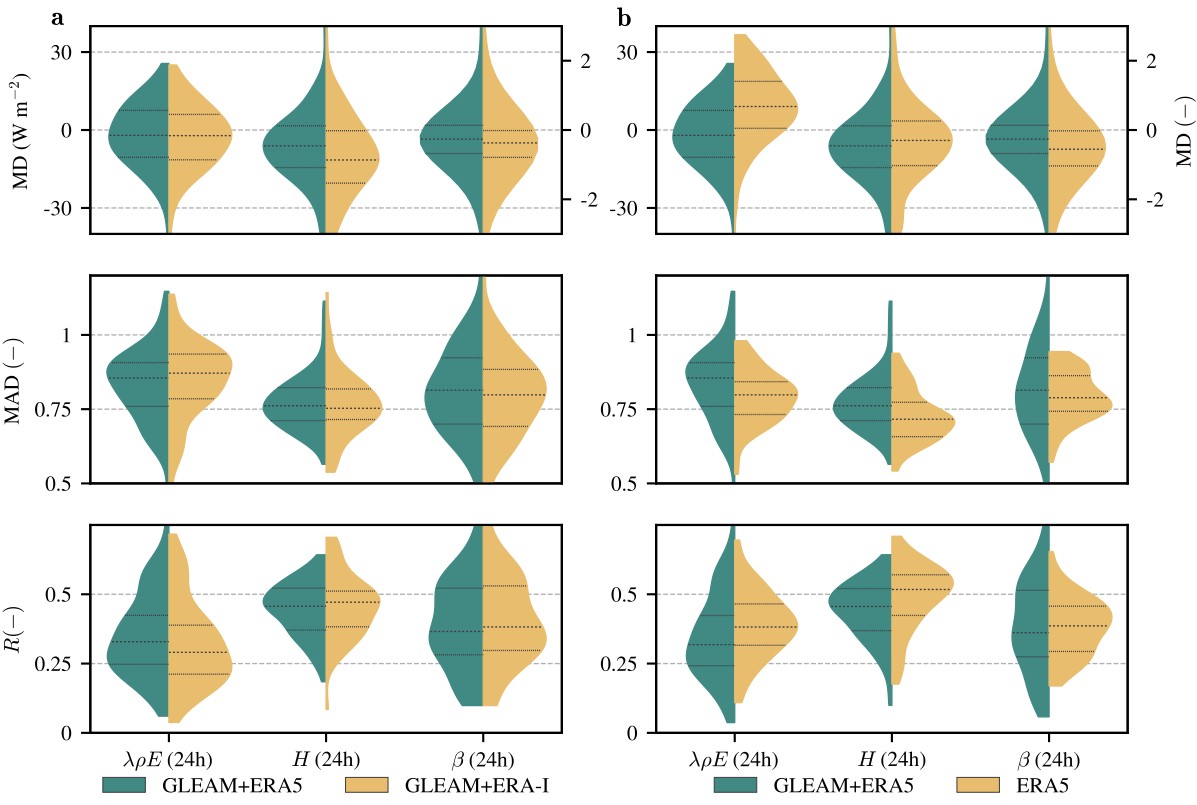

**Figure 4.** Violin plots of temporal validation statistics of the surface latent heat flux ($\lambda\rho E$), surface sensible heat flux ($H$), and Bowen ratio ($\beta$) from GLEAM+ERA5, GLEAM+ERA-I, and ERA5. **(a)** Compares the violin plots from GLEAM+ERA5 and GLEAM+ERA-I and **(b)** compares the violin plots from GLEAM+ERA5 and ERA5. Statistics are calculated against in situ eddy-covariance measurements at daily temporal resolution. Violin plots represent the distribution of the individual validation statistics with indication of the median and inter-quartile range, and are calculated using a kernel density estimation approach. Statistics include the Mean Difference (MD, raw in situ time series from 143 sites as reference), Mean Absolute Difference (MAD, anomaly in situ time series from 77 sites as reference), and the Pearson correlation coefficient ($R$, anomaly in situ time series from 77 sites as reference). The distribution of the MD of $\beta$ is plotted on the right y-axis.

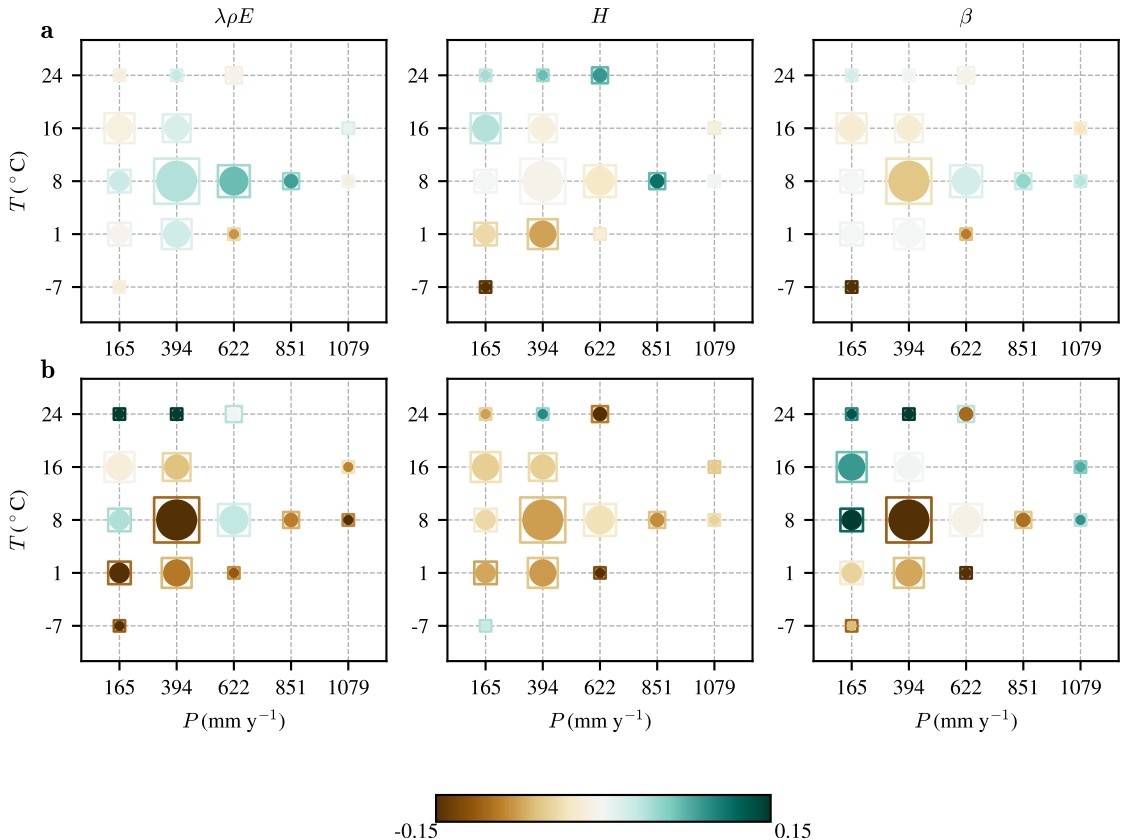

**Figure 5.** Difference between temporal validation statistics of the surface latent heat flux ($\lambda\rho E$), surface sensible heat flux ($H$), and Bowen ratio ($\beta$) from GLEAM+ERA5, GLEAM+ERA-I, and ERA5 grouped as a function of precipitation rate ($P$) and near-surface air temperature ($T$) calculated at the in situ site. **(a)** Compares the statistics from GLEAM+ERA5 and GLEAM+ERA-I and **(b)** compares the statistics from GLEAM+ERA5 and ERA5. Statistics are calculated against in situ eddy-covariance measurements at daily resolution and then averaged across the sites within each group. Statistics include the Mean Absolute Difference (MAD, anomaly in situ time series from 77 sites as reference) and the Pearson correlation coefficient ($R$, anomaly in situ time series from 77 sites as reference). In **(a)** circles show the $R$ from GLEAM+ERA5 minus the one from GLEAM+ERA-I, while squares show the MAD from GLEAM+ERA-I minus the one from GLEAM+ERA5; hence, green colors represent better statistics for GLEAM+ERA5 compared to GLEAM+ERA-I. In **(b)**, statistics from GLEAM+ERA-I are replaced by ERA5; hence, green colors represent better statistics for GLEAM+ERA5 compared to ERA5. The size of the symbols relates to the number of in situ sites per group.

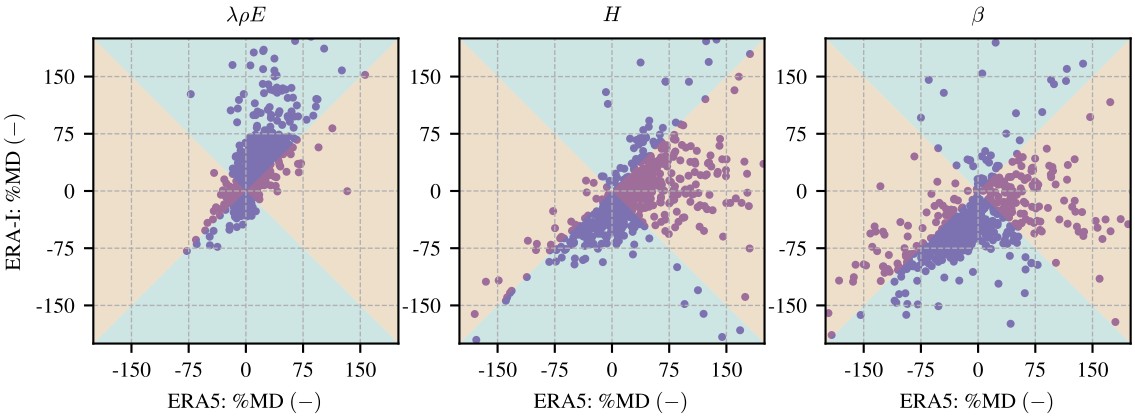

**Figure 6.** Scatter plot of the bias of the surface latent heat flux ($\lambda\rho E$), surface sensible heat flux ($H$), and Bowen ratio ($\beta$) from ERA-I versus ERA5. The bias is calculated against catchment-scale estimates of the fluxes derived using discharge data (Eqs. 1–3) and is assessed by the percentage Mean Difference (%MD, raw time series from 707 catchments as reference). The green area indicates points where the bias in ERA5 is better than in ERA-I, and vice versa for the brown area.

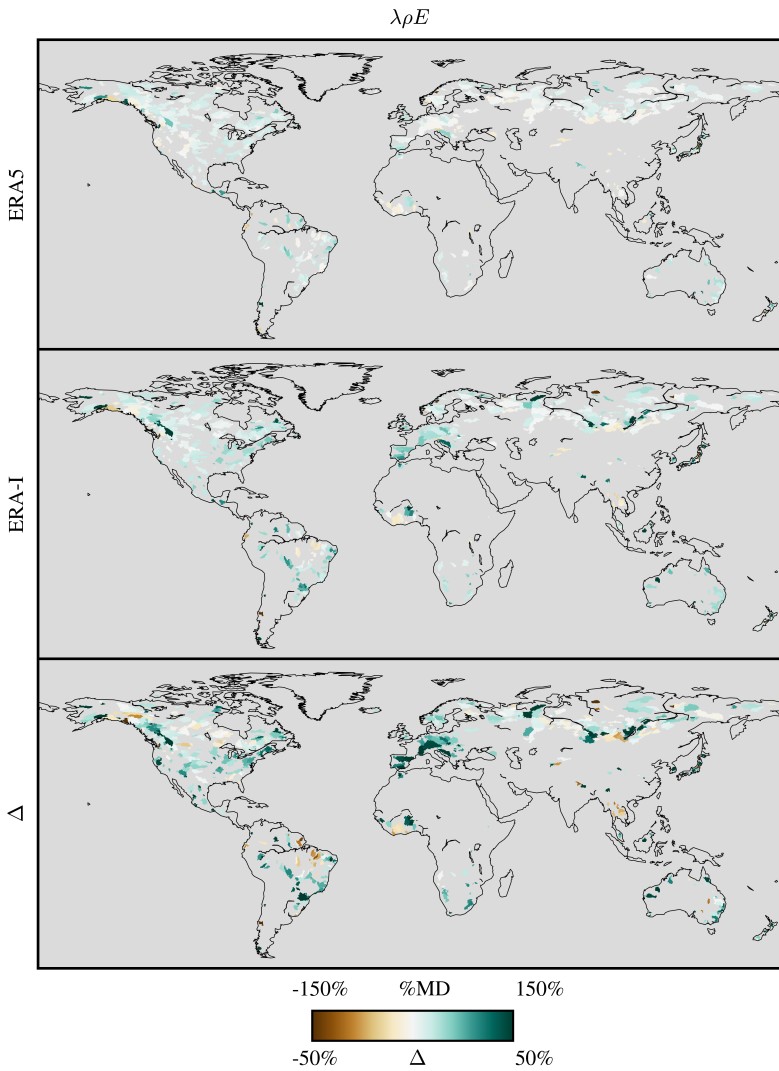

**Figure 7.** Maps of the bias of the surface latent heat flux ($\lambda \rho E$) from ERA5 and ERA-I. The bias is calculated against catchment-scale estimates of the fluxes derived using discharge data (Eqs. 1–3) and is assessed by the percentage Mean Difference (%MD, raw time series from 707 catchments as reference). The bottom map represents the difference ($\Delta$) between the absolute bias in ERA-I and ERA5; hence, green colors represent lower bias in ERA5 than in ERA-I.

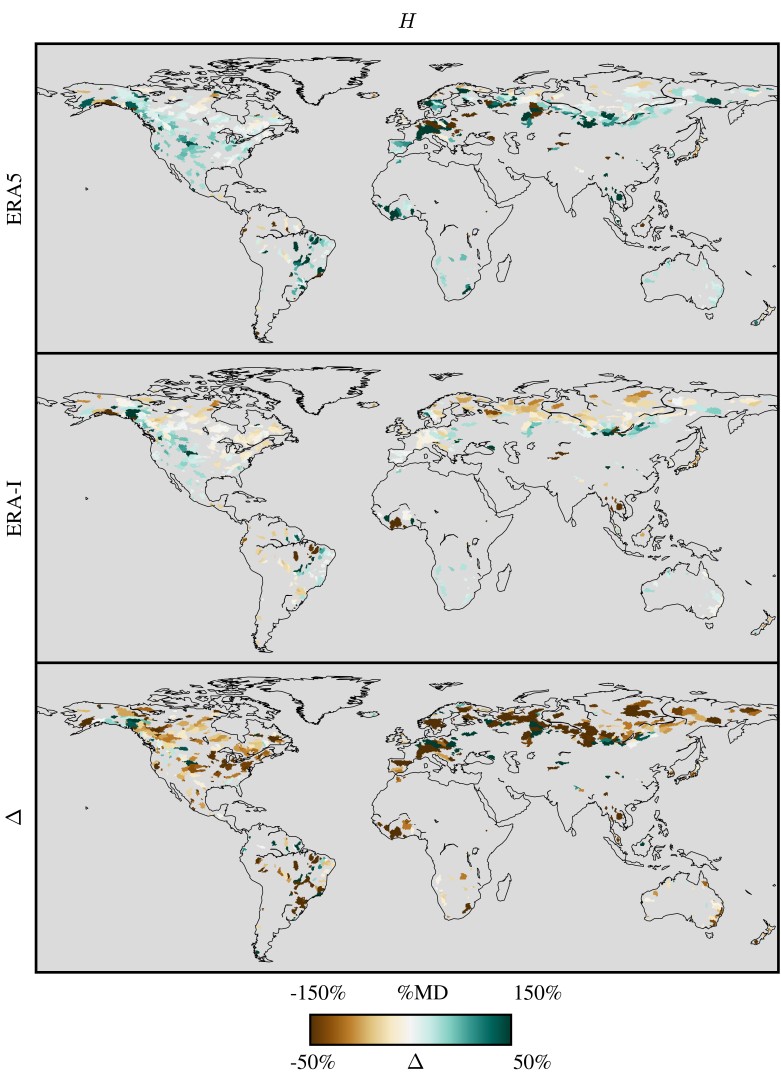

**Figure 8.** Like Fig. 7, but for the surface sensible heat flux ($H$).

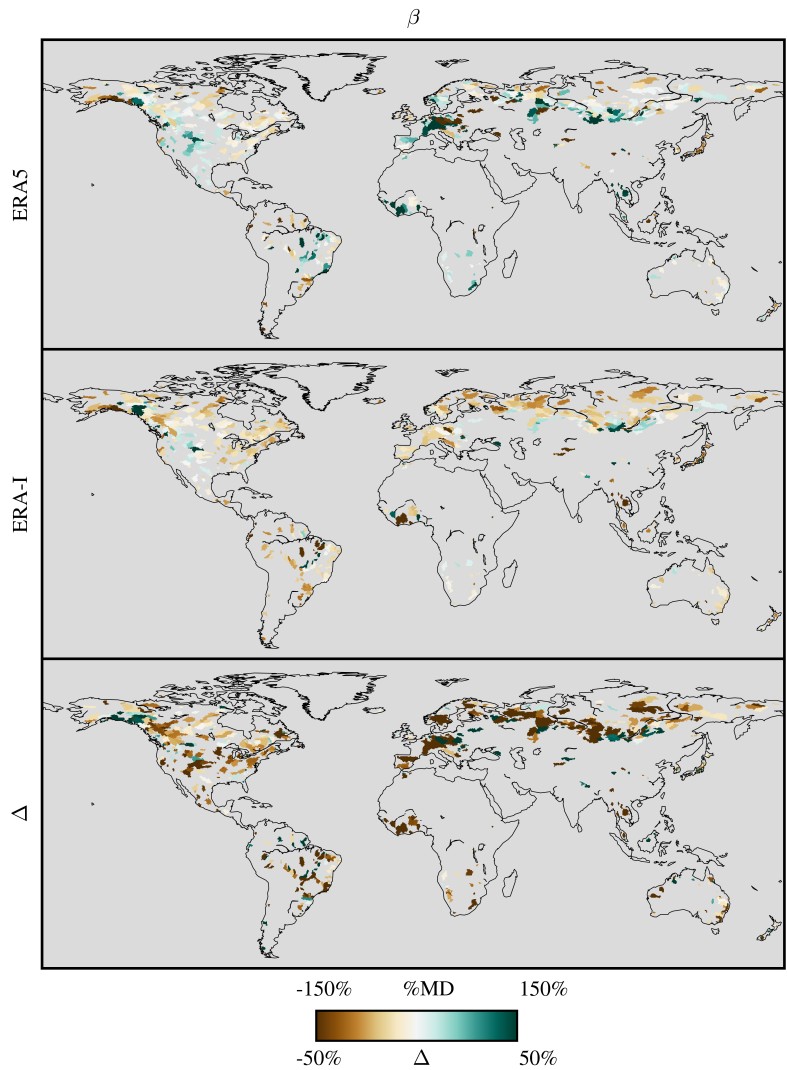

**Figure 9.** Like Fig. 7, but for the Bowen ratio ($\beta$).

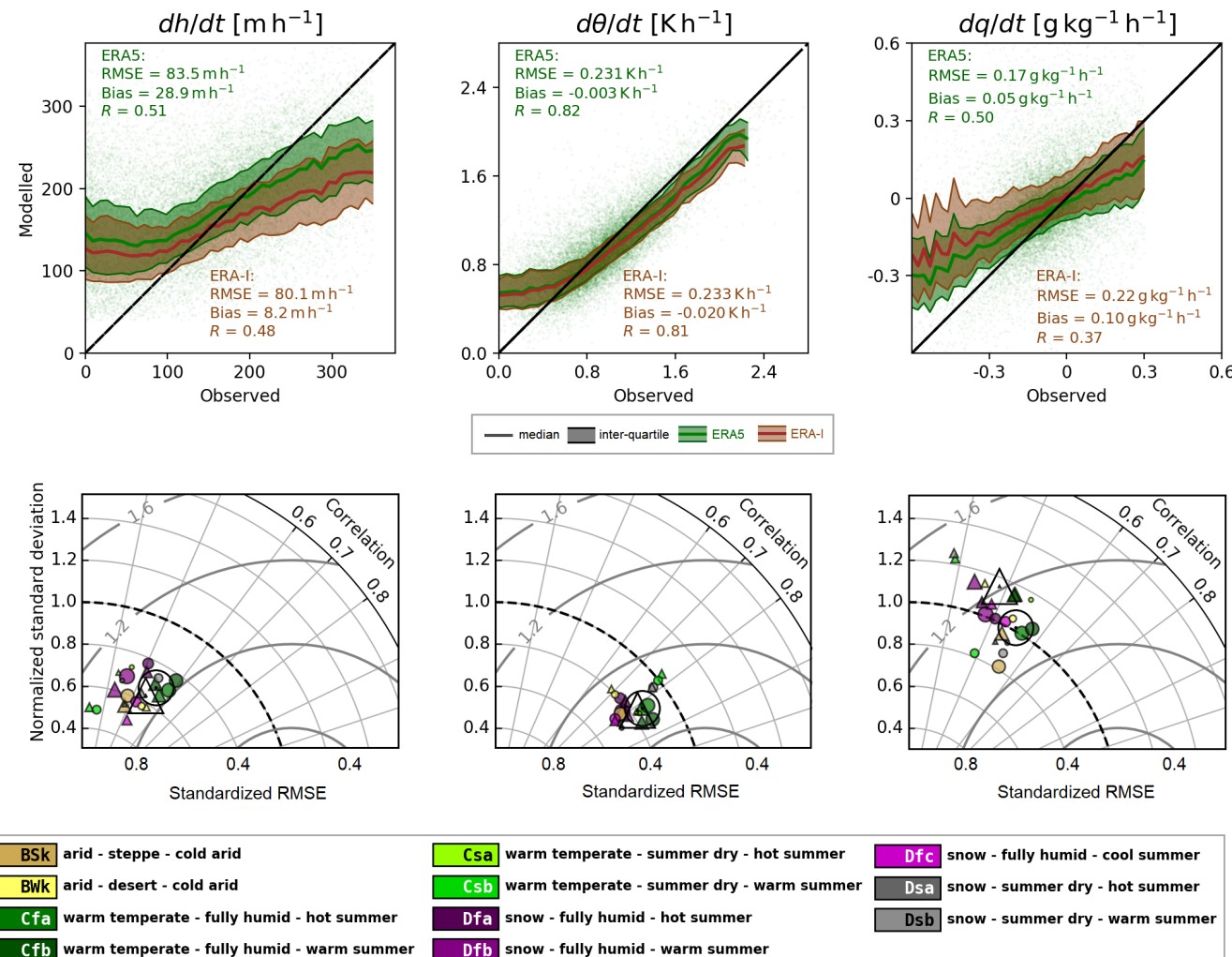

**Figure 10.** Skill of CLASS4GL at reproducing diurnal changes in ABL properties when forced with surface evaporative fractions from ERA5 versus ERA-I. Shown are the tendencies of the mixed-layer height ($dh/dt$), potential temperature ($d\theta/dt$), and specific humidity ($dq/dt$), which are assessed by comparison of model simulations against the IGRA sounding data between 1981 and 2015. The first row shows modeled versus observed data points, and the corresponding median and inter-quartile range of the simulations in solid lines, where green represents ERA5 and brown ERA-I. The 1–1 line is shown as a black line for reference. The bottom row illustrates the skill of the ABL simulations when forced with ERA5 (circles) versus ERA-I (triangles) in the form of Taylor plots. The transparent symbols show the overall performance of 18000 sounding pairs from 121 stations, whereas the colored symbols indicate the performance per Köppen-Geiger climate class and for which the size is proportional to the number of sounding pairs.

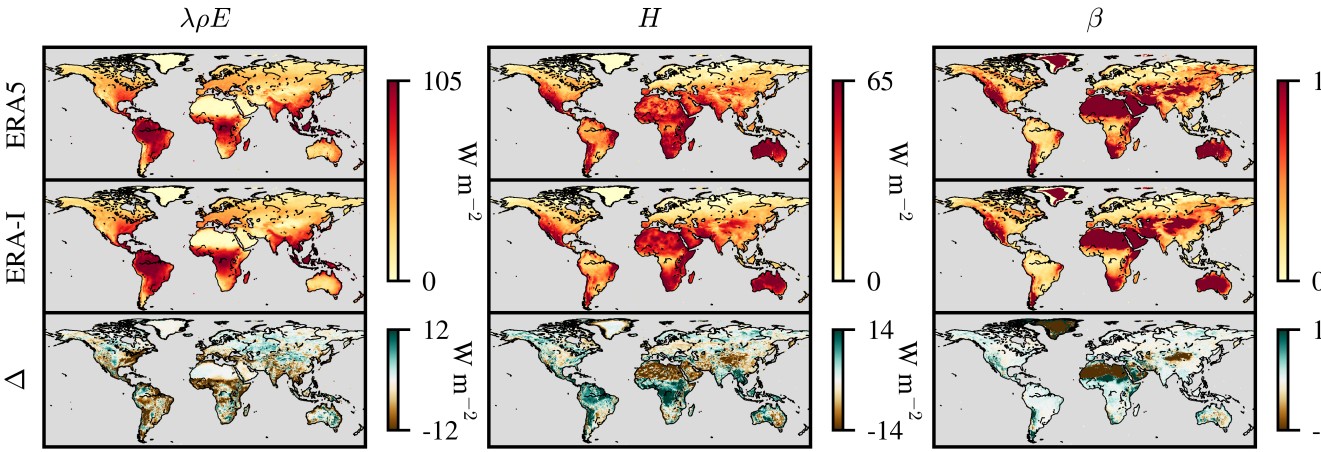

**Figure 11.** Maps of the multi-annual average of surface latent heat flux ($\lambda \rho E$, W m$^{-2}$), surface sensible heat flux ($H$, W m$^{-2}$), and Bowen ratio ($\beta$) from ERA5 and ERA-I. In the last row, $\Delta$ presents the difference between ERA5 and ERA-I; hence, green colors represent higher values in ERA5 compared to ERA-I.

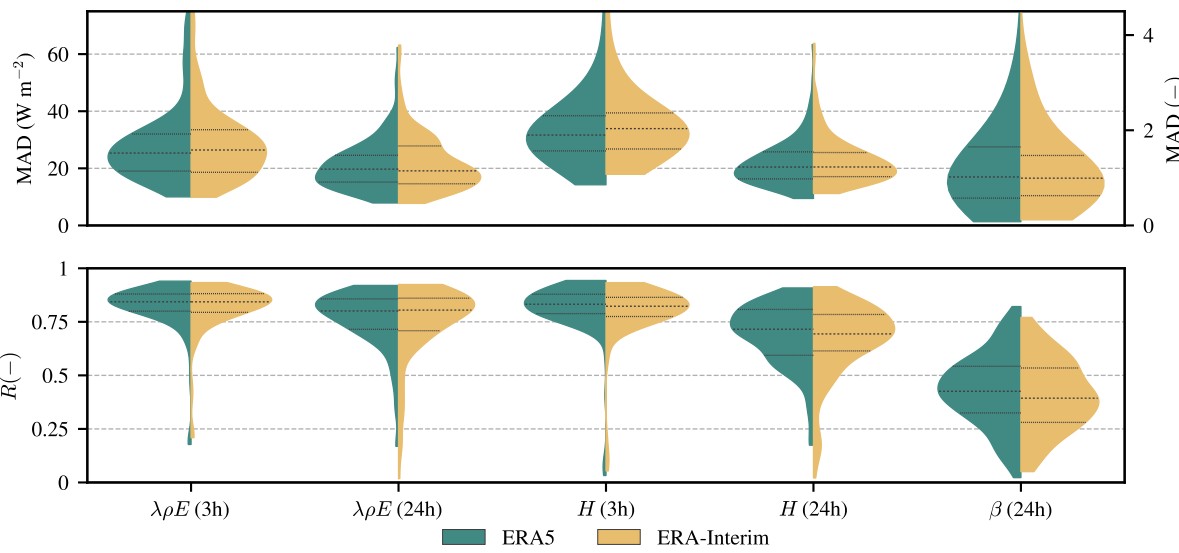

**Figure A1.** Violin plots of temporal validation statistics of the surface latent heat flux ($\lambda\rho E$), surface sensible heat flux ($H$), and Bowen ratio ($\beta$) from ERA5 (green) and ERA-I (yellow). Statistics are calculated against in situ eddy-covariance measurements at both 3-hourly and daily temporal resolutions. Violin plots represent the distribution of the individual validation statistics with indication of the median and inter-quartile range, and are calculated using a kernel density estimation approach. Statistics include the Mean Absolute Difference (MAD, raw in situ time series from 143 sites as reference) and the Pearson correlation coefficient ($R$, raw in situ time series from 143 sites as reference). The distribution of the MAD of $\beta$ is plotted on the right y-axis.

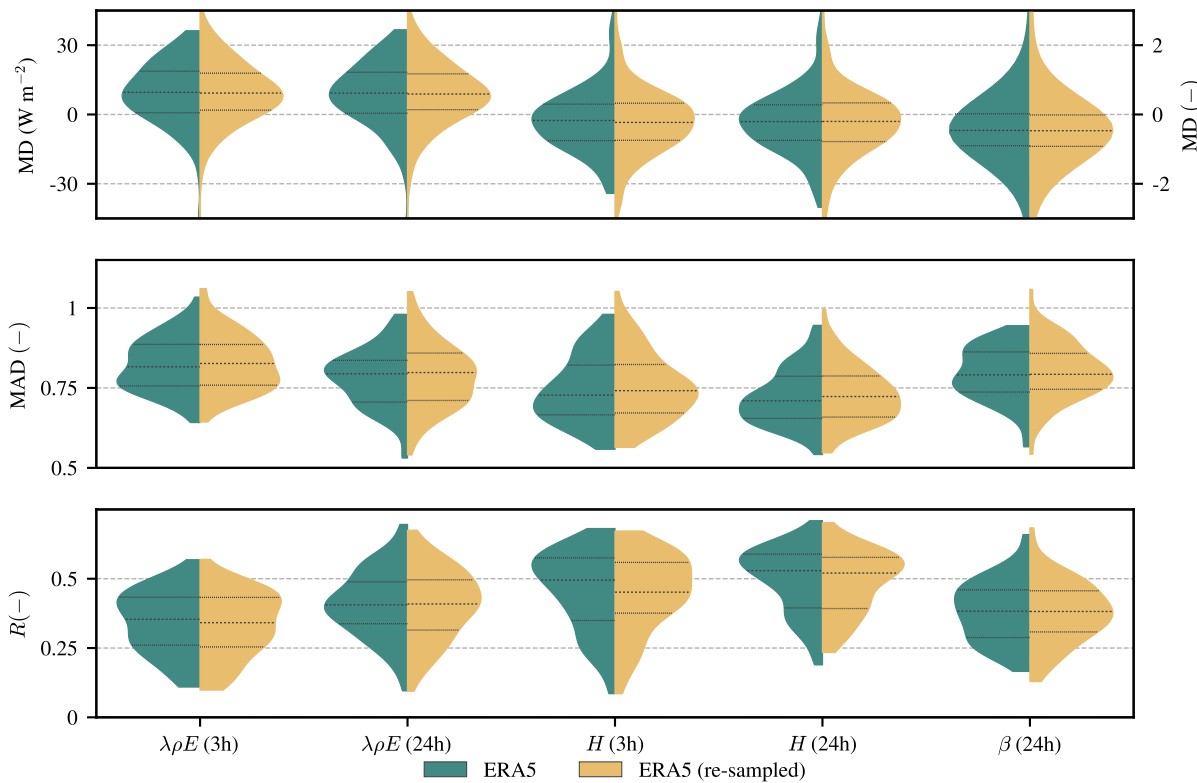

**Figure A2.** Violin plots of temporal validation statistics of the surface latent heat flux ($\lambda\rho E$), surface sensible heat flux ($H$), and Bowen ratio ($\beta$) from ERA5 (green) and ERA5 linearly re-sampled to the spatial grid of ERA-I (yellow). Statistics are calculated against in situ eddy-covariance measurements at both 3-hourly and daily temporal resolutions. Violin plots represent the distribution of the individual validation statistics with indication of the median and inter-quartile range and are calculated using a kernel density estimation approach. Statistics include the Mean Difference (MD, raw in situ time series from 143 sites as reference), Mean Absolute Difference (MAD, anomaly in situ time series from 77 sites as reference), and the Pearson correlation coefficient ($R$, anomaly in situ time series from 77 sites as reference). The distribution of the MD of $\beta$ is plotted on the right y-axis.

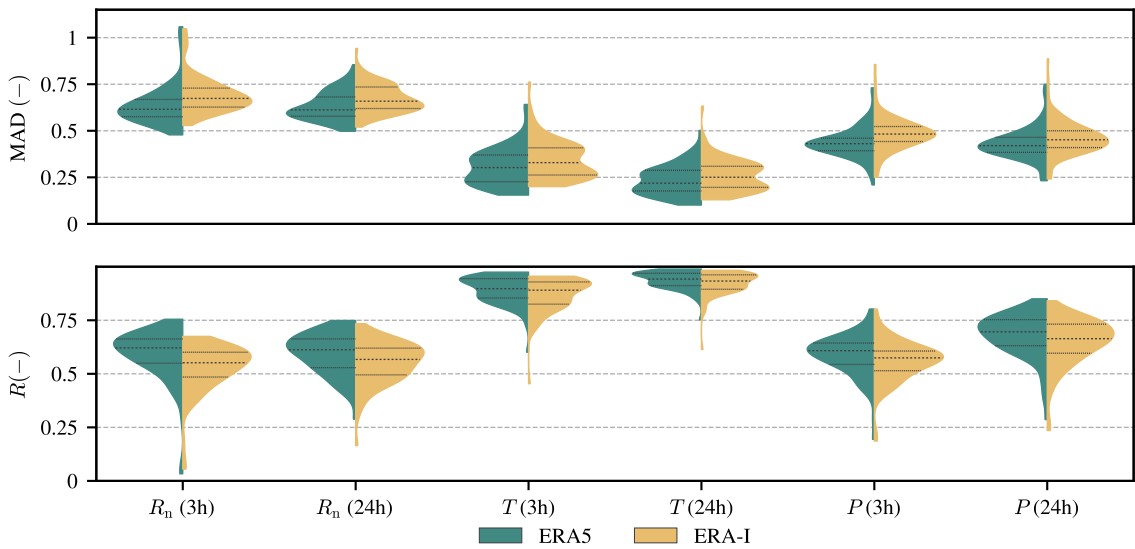

**Figure A3.** Violin plots of temporal validation statistics of the surface net radiation ($R_n$), 2-meter air temperature ($T$), and precipitation rate ($P$) from ERA5 (green) and ERA-I (yellow). Statistics are calculated against in situ eddy-covariance measurements at both 3-hourly and daily temporal resolutions. Violin plots represent the distribution of the individual validation statistics with indication of the median and inter-quartile range, and are calculated using a kernel density estimation approach. Statistics include the Mean Absolute Difference (MAD, anomaly in situ time series from 83 sites as reference) and the Pearson correlation coefficient ($R$, anomaly in situ time series from 83 sites as reference).

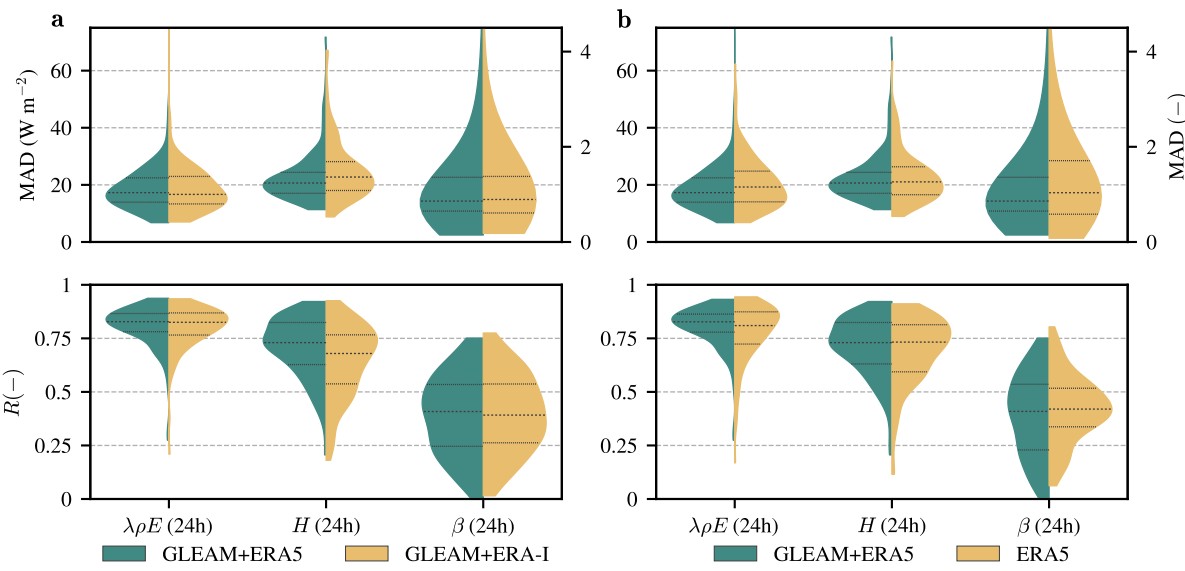

**Figure A4.** Violin plots of temporal validation statistics of the surface latent heat flux ($\lambda\rho E$), surface sensible heat flux ($H$), and Bowen ratio ($\beta$) from GLEAM+ERA5, GLEAM+ERA-I, and ERA5. **(a)** Compares the violin plots from GLEAM+ERA5 and GLEAM+ERA-I and **(b)** directly compares the violin plots from GLEAM+ERA5 and ERA5. Statistics are calculated against in situ eddy-covariance measurements at daily temporal resolution. Violin plots represent the distribution of the individual validation statistics with indication of the median and inter-quartile range and are calculated using a kernel density estimation approach. Statistics include the Mean Absolute Difference (MAD, raw in situ time series from 143 sites as reference) and the Pearson correlation coefficient ($R$, raw in situ time series from 143 sites as reference). The distribution of the MAD of $\beta$ is plotted on the right y-axis.

*Author contributions.* B.M. and D.G.M. conceived the study and designed the lay-out. B.M. processed the reanalyses data, eddy-covariance data, and performed the simulations with GLEAM. D.S. processed the GRDC discharge data. H.W. processed the balloon sounding data and performed the CLASS4GL simulations. B.M. and D.G.M. designed the lay-out of the manuscript and B.M. led the writing. All authors have been involved in interpreting the results, discussing the findings, and editing the manuscript.

5  *Competing interests.* The authors declare no competing interests.

*Acknowledgements.* This work was partly funded by the Belgian Science Policy Office through the ET-Sense project (SR/02/377). D.G.M., H.W., and D.S. acknowledge support from the European Research Council (ERC) under grant agreement n° 715254 (DRY-2-DRY). This work used eddy-covariance data acquired and shared by the FLUXNET community, including these networks: AmeriFlux, AfriFlux, Asi-aFlux, CarboAfrica, CarboEuropeIP, CarboItaly, CarboMont, ChinaFlux, Fluxnet-Canada, GreenGrass, ICOS, KoFlux, LBA, NECC, OzFlux-
10  TERN, TCOS-Siberia, and USCCC. The FLUXNET eddy covariance data processing and harmonization was carried out by the ICOS Ecosystem Thematic Center, AmeriFlux Management Project and Fluxdata project of FLUXNET, with the support of CDIAC, and the OzFlux, ChinaFlux and AsiaFlux offices. The authors also acknowledge Instituto Nacional Technologica Agropecuaria for making the eddy-covariance data of AR-Vir publically available.

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
