# Peer review of "Evaluating the surface energy partitioning in ERA5"

_Geoscientific Model Development, 2019_

## Referee Comment (RC1) · Anonymous Referee #1 · 11 Mar 2020

The paper investigates the skill of the land surface energy partitioning in the ERA5 reanalysis. For reference, the ERA5 skill is compared to that of its predecessor, ERA-Interim (ERA-I). Skill is determined in several different ways: (1) directly vs flux tower measurements, (2) vs fluxes from water and energy balance estimates, (3) by driving the GLEAM land surface scheme with reanalysis data and validating the output vs. flux tower measurements, and, finally, (4) by driving the CLASS4GL boundary layer model with reanalysis data and validating the output vs. balloon observations.

The authors find that ERA5 land surface energy partitioning is generally improved over that of ERA-I. In particular, the overestimation of the latent heat flux in ERA-I is reduced (but not eliminated) in ERA5.

The paper is of interest to GMD readers and makes an important contribution by doc-

umenting the quality of ERA5 land surface estimates. By and large, the writing and graphics are clear and concise, and the conclusions drawn from the study are supported by the results. I recommend eventual publication of the paper in GMD provided the authors address the comments below. It would be particularly helpful to include other reanalysis data in the comparison.

Major comments (no particular order):

1) The title and introduction do not make it sufficiently clear that the turbulent fluxes investigated here are for the land only. I suggest changing the title to: "Evaluating the *land* surface energy partitioning in ERA5" and occasionally replacing the expression "surface [..] fluxes" with "land surface [..] fluxes", e.g., P3/L15, P4/L3, and probably a few more places.

2) P4/L1: Aren't there advances in land data assimilation from ERA-I to ERA5? And do these not matter for the quality of the land surface turbulent flux estimates? Land data assimilation in ERA-I and ERA5 and its impact on land surface estimates should be included briefly in the Introduction and further discussed in the Results and Discussion section.

3) Section 2 is a somewhat odd mix of "methods" and "data". E.g., the FLUXNET 2015 section 2.2 includes discussion of how the climatology is derived for the computation of the standardised anomalies, but this would also apply to the other datasets (incl. ERA-I and ERA5). The entire section needs to be reorganized and be more clearly separated into "Data" and "Methods".

4) The violin plots are great visual tools, but I assume their construction involves is some fitting of the distributions. These details should be in the "Methods" section.

5) There are gaps in the literature discussion. E.g., Draper et al (2018) is a highly relevant assessment of reanalysis estimates of land surface energy flux estimates, incl ERA-I. Draper, C. S., R. H. Reichle, and R. D. Koster (2018), Assessment of

MERRA-2 Land Surface Energy Flux Estimates, Journal of Climate, 31, 671-691, doi:10.1175/JCLI-D-17-0121.1.

6) Related to 5), the present paper only investigates ERA-I and ERA5. The paper would be considerably more relevant to readers if these ECMWF products were assessed along with at least one or two other, major reanalysis products (such as MERRA-2).

7) If I understood this correctly, the GLEAM and CLASS4GL analyses work as follows: (i) use ERA-I and, separately, ERA5 to force GLEAM (or CLASS4GL), then (ii) evaluate the results against tower (or balloon) measurements. This approach depends on the assumption that GLEAM and CLASS4GL are very good models, or at least that they do not have compensating errors. If, say, errors in GLEAM were to compensate for errors in ERA-I, forcing GLEAM with a better reanalysis isn't necessarily going to deliver better GLEAM outputs. Similarly for CLASS4GL. This is a major caveat that needs to be discussed prominently in the paper.

8) Figures A.1-A.4 are oddly placed. Either they need to be put into a proper Appendix, with discussion in the Appendix as well, or they need to placed in a separate "Supplementary Information" document. As assembled, Figures A.1-A.4 are simply out of order.

9) Figures 2 and 4 are a confusing mix of dimensional metrics (MD for raw fluxes) and unit-less metrics (MD for raw Bowen ratio, and MAD and R for standardised anomalies). These figures also lack basic information about a dimensional 2nd-order metric such as MAD or RMSE for dimensional (raw or anomaly) variables. The readers are going to want to know typical MAD or RMSE values for fluxes in units of W/m2, incl. and/or excl. the seasonal cycle.

I suggest revisiting the assembly of Figures 2, 4, A.1-A.4. The paper would be much easier to follow and more informative if, say, one figure includes only (dimensional) metrics computed from raw time series and another figure includes only (non-dimensional) metrics from standardised anomaly time series.

10) P10/L15-16: "Figure 4b shows that ERA5 is better at capturing..." Doesn't this invalidate the conclusion drawn from Fig 4a, that is, the evaluation of ERA-I and ERA5 through GLEAM? See also comment 7) above.

11) P11/L3-8: One aspect that may come into play here is that GLEAM+ERA5 is an off-line (land-only) modeling system that does not permit feedback whereas ERA5 is a coupled land-atmosphere modeling system. This may be related to a finding by Draper et al (2018):

"Finally, the SH results for MERRA-Land are troubling. While MERRA-Land did have the desired reduction in the LH biases compared to MERRA (to 1 W/m2 in the global land annual average), it also had a compensating, and much larger, increase in the SH bias (up to 15W m22 in the global land average)" [beginning of p689 of Draper et al. 2018]

See also comment 5) above about the need for a better integration of the results of the present paper into the literature context.

12) P11/L26-P12/ L2: There is no discussion of Fig 9! In this paragraph, insert explicit references to Figs 7, 8, and 9 in the relevant place within the paragraph. E.g., reference Fig 7 in P11/L28, reference Fig 8 in P11/L33. This reveals that Fig 9 has not been discussed.

13) In some cases the results are overstated.

E.g., P12/L22: "The improvements are less clear..." suggests that there are some (hard-to-see) improvements, when in fact the results are neutral at best.

P14/L3-4: The statement here is not consistent with the results of Fig 4 that show that ERA-I estimates of the sensible heat flux and Bowen ratio are better than those of ERA5.

14) The tower validation results should come with some measure of statistical significance or error bars. Are the improvements, that is, the small shifts in the distributions

of the metrics as shown in the violin plots, meaningful?

Minor comments:

a) P2/L13: Reichle et al. 2017 is a reference primarily for MERRA-2 land surface estimates. MERRA-2 which is a full atmospheric reanalysis, similar to ERA-I and ERA5. In this place, however, the authors are here referring to land-only reanalyses, such as ERA-Interim/Land and MERRA-Land. For the latter, Reichle et al. (2011) is a better reference. Note that there is *not* a land-only reanalysis associated with MERRA-2.

Reichle, R. H., et al. (2011), Assessment and enhancement of MERRA land surface hydrology estimates, Journal of Climate, 24, 6322-6338, doi:10.1175/JCLI-D-10-05033.1.

b) P2/L26-30: The text here is about very geographically limited results (southern Antarctic peninsula) or sea-ice, which is not the focus of the present paper. I suggest deleting this text or moving it further down. It confuses the reader by distracting from the focus of the paper on the global land surface turbulent fluxes.

c) P4/L19: On first reading, I completely missed the term "standardised" here and later got confused about the lack of dimensions/units in the graphics. In many papers, anomalies from the seasonal cycle are examined without standardisation, i.e., they are dimensional anomalies. There is nothing wrong per se with the standardised anomalies, but please make it clearer that you are focusing on dimensionless anomalies.

d) P10/L31: typo: "we should emphasis" –> "we should emphasise"

e) P12/L16+L20: The numbers referenced here contradict the numbers in the graphic.

f) Caption Fig 2: Replace "For MD, the distribution of beta..." with "The distribution of the MD of beta..."???

g) Caption Fig 7: "...between the absolute bias in ERA5 and ERA-I;"??? Maybe I'm misunderstanding this, but I think the bottom panel shows abs(bias(ERA-I)) minus

abs(bias(ERA5), that is, the sign of the abs bias difference is different from what the caption suggests.

h) Caption Fig 10: "...versus ERA-I (squares)"??? Should this read "...versus ERA-I (triangles)"???

i) Figure 11 needs units on the colorbars. I also suggest making the graphic bigger so it can be read more clearly in a hard copy for the next round of reviews.

---

## Referee Comment (RC2) · Anonymous Referee #2 · 24 Mar 2020

General comments:

This is a very interesting and useful study of the representation of the land surface energy budget in European global atmospheric reanalyses. A large number of diverse in situ observations are used to benchmark several simulations at a global scale. Overall, the paper is well written, apart from the mixing of results and discussion/interpretation. Quality of some Figures could be improved. Colour scales are sometimes confusing as "green" tends to look blue. Violin plots are useful but do not provide a point by point comparison. Could all the corresponding scatter plots be given in a Supplement? A discussion on the impact of land cover is lacking.

Recommendation: minor revisions.

[Figure]

Particular comments:

- P. 1, Title: should be more specific. For example: "Evaluating the land surface energy partitioning in European global atmospheric reanalyses".

- P. 3, L. 1-2 ("perform better than ERA5"): any reference on this?

- P. 4, L.7: I would be more specific. For example: "the more evolved HTESSEL land surface model in ERA5".

- P. 4, L. 19: could you explain how these anomalies are defined and calculated?

- P. 5, L. 1-3: It seems that a key issue was not addressed. Land cover type in ERA5 may not correspond to the tower's one. E.g. a grassland Fluxnet site may be located in an ERA5 grid cell mainly covered by forests. How did you handle this?

- P. 6, L. 5: could you define "non-overlapping moving windows"?

- P. 6, L. 12 (G as a fixed fraction of Rn): Could this explain the poor scores obtained for sensible heat flux in Figure 8? The soil heat flux is related to soil properties and can be influenced by sensible heat exchange with rainwater (e.g. Zhang et al. https://doi.org/10.5194/acp-19-5005-2019).

- P. 6, L. 12 ("land cover"): which land cover? Is it the land cover used in the model?

- P. 8, L. 3: Is there an impact of the land cover type?

- P. 8, L. 17: Seasonality removal should be described is chapter 2.

- P. 8, L. 28: what about Fluxnet site distribution in terms of vegetation types?

Editorial comments (Figures):

- Figure 1: Sites cannot be easily spotted. Colors of dots and background should be changed. What about land cover types? Format of the subfigure on the right should be consistent with format of Figure 3.

- Figure 6: green or blue?

- Figure 10 (top subfigures): meaning of the red lines? These metrics are a bit obscure. Why not comparing scatterplots of ABL heights?

- Figure 11: Not readable. Difference figures should be expanded. Green or blue?
* * *

---

## Author Comment (AC1) · 24 Apr 2020

RC1

The paper investigates the skill of the land surface energy partitioning in the ERA5 reanalysis. For reference, the ERA5 skill is compared to that of its predecessor, ERA Interim (ERA-I). Skill is determined in several different ways: (1) directly vs flux tower measurements, (2) vs fluxes from water and energy balance estimates, (3) by driving the GLEAM land surface scheme with reanalysis data and validating the output vs. flux tower measurements, and, finally, (4) by driving the CLASS4GL boundary layer model with reanalysis data and validating the output vs. balloon observations. The authors find that ERA5 land surface energy partitioning is generally improved over that of ERA-I. In particular, the overestimation of the latent heat flux in ERA-I is reduced (but not eliminated) in ERA5.

The paper is of interest to GMD readers and makes an important contribution by documenting the quality of ERA5 land surface estimates. By and large, the writing and graphics are clear and concise, and the conclusions drawn from the study are supported by the results. I recommend eventual publication of the paper in GMD provided the authors address the comments below. It would be particularly helpful to include other reanalysis data in the comparison.

**We thank the referee for reviewing the paper and providing us with useful comments, feedback, and corrections. Below, we give a point-to-point reply to the comments posted by the reviewer and list the changes that will be implemented in the revised version of the manuscript.**

**Major comments (no particular order):**

1. The title and introduction do not make it sufficiently clear that the turbulent fluxes investigated here are for the land only. I suggest changing the title to: "Evaluating the \*land\* surface energy partitioning in ERA5" and occasionally replacing the expression "surface [..] fluxes" with "land surface [..] fluxes", e.g., P3/L15, P4/L3, and probably a few more places.
   **The authors agree with the referee and will update the title.**

   **Changes in manuscript: the title of the manuscript will be updated per suggestion of the referee.**

2. P4/L1: Aren't there advances in land data assimilation from ERA-I to ERA5? And do these not matter for the quality of the land surface turbulent flux estimates? Land data assimilation in ERA-I and ERA5 and its impact on land surface estimates should be included briefly in the Introduction and further discussed in the Results and Discussion section.
   **The authors agree with the referee that changes in the data assimilation system were not well-emphasized in the manuscript.**

   **Changes in manuscript: the changes in the data assimilation system will be more highlighted in the revised version of the manuscript.**

3. Section 2 is a somewhat odd mix of "methods" and "data". E.g., the FLUXNET 2015 section 2.2 includes discussion of how the climatology is derived for the computation of the standardised anomalies, but this would also apply to the other datasets (incl. ERA-I and ERA5). The entire section needs to be reorganized and be more clearly separated into "Data" and "Methods".
   **The authors agree that this would improve the readability of the manuscript.**

   **Changes in the manuscript: the authors will re-organise Section 2 and introduce a section where methods (such as metrics calculation and anomaly calculation) are described.**

4.  The violin plots are great visual tools, but I assume their construction involves is some fitting of the distributions. These details should be in the "Methods" section.
    **The method used to construct the violin plots is mentioned in the caption of the figures and involves a kernel density estimation approach.**

    **Changes in the manuscript:** the authors will briefly mention this in the method section of the revised manuscript.

5.  There are gaps in the literature discussion. E.g., Draper et al (2018) is a highly relevant assessment of reanalysis estimates of land surface energy flux estimates, incl ERA-I. Draper, C. S., R. H. Reichle, and R. D. Koster (2018), Assessment of MERRA-2 Land Surface Energy Flux Estimates, Journal of Climate, 31, 671-691, doi:10.1175/JCLI-D-17-0121.1.
    **The authors thank the reviewer for this suggestion.**

    **Changes in the manuscript:** the authors will cite the paper in the manuscript, and reconsider whether other references may be of relevance.

6.  Related to (5), the present paper only investigates ERA-I and ERA5. The paper would be considerably more relevant to readers if these ECMWF products were assessed along with at least one or two other, major reanalysis products (such as MERRA-2).
    **The authors thank the referee for this suggestion and fully agree that this would be an interesting analysis. However, the focus of the paper is to make a first assessment of the quality of the turbulent fluxes in ERA5 against different observation-based data sets. Including other reanalyses – next to ERA-I, which is there to show improvements of ERA5 upon its predecessor – would deviate the focus from ERA5, and turn this manuscript into a reanalysis comparison paper.**

    **Changes in the manuscript:** no changes.

7.  If I understood this correctly, the GLEAM and CLASS4GL analyses work as follows: (i) use ERA-I and, separately, ERA5 to force GLEAM (or CLASS4GL), then (ii) evaluate the results against tower (or balloon) measurements. This approach depends on the assumption that GLEAM and CLASS4GL are very good models, or at least that they do not have compensating errors. If, say, errors in GLEAM were to compensate for errors in ERA-I, forcing GLEAM with a better reanalysis isn't necessarily going to deliver better GLEAM outputs. Similarly for CLASS4GL. This is a major caveat that needs to be discussed prominently in the paper.
    **The authors agree that this type of analysis comes with assumptions, yet it still provides useful insights in the quality of ERA5. Unless errors in ERA5 and GLEAM (or CLASS4GL) are somehow (anti-)correlated, forcing either of the models with better inputs can only lead to better model output as compared against in situ observations when done systematically at a large number of sites. Given the strong differences between the model concepts of ERA5 and GLEAM (or CLASS4GL), the authors believe that both are sufficiently independent to be used in such analysis.**

    **Changes in the manuscript:** this issue will be briefly discussed in the revised version of the manuscript.

8. Figures A.1-A.4 are oddly placed. Either they need to be put into a proper Appendix, with discussion in the appendix as well, or they need to placed in a separate "Supplementary Information" document. As assembled, Figures A.1-A.4 are simply out of order.
**The figures will be uploaded as supplementary material, following the guidelines of Copernicus.**

**Changes in the manuscript:** the figures will be uploaded as supplementary material, following the guidelines of Copernicus.

9. Figures 2 and 4 are a confusing mix of dimensional metrics (MD for raw fluxes) and unit-less metrics (MD for raw Bowen ratio, and MAD and R for standardised anomalies). These figures also lack basic information about a dimensional 2nd-order metric such as MAD or RMSE for dimensional (raw or anomaly) variables. The readers are going to want to know typical MAD or RMSE values for fluxes in units of W/m2, incl. and/or excl. the seasonal cycle. I suggest revisiting the assembly of Figures 2, 4, A.1-A.4. The paper would be much easier to follow and more informative if, say, one figure includes only (dimensional) metrics computed from raw time series and another figure includes only (non-dimensional) metrics from standardised anomaly time series.
**The primary reason to focus on the evaluation of anomalies is to minimise the effect of the strong seasonal cycle in the turbulent fluxes, which might mask important differences between the quality of ERA5 and ERA-I. Then, we prefer to focus on standardised anomalies to allow direct comparison of metrics from the different turbulent fluxes that typically range in a different order of magnitude. Needless to say that calculating the Mean Difference on anomaly time series is useless as anomalies are mean zero; hence we report the Mean Difference calculated on raw time series.**

**However, we do agree with the referee that readers might be interested in the statistics calculated on raw time series as well. Therefore, we report these corresponding statistics in Figures A1 and A4.**

**Changes in the manuscript:** The authors believe that the content of the figures is described with sufficient detail in the captions and corresponding text to avoid confusion, thus we foresee no changes.

10. P10/L15-16: "Figure 4b shows that ERA5 is better at capturing..." Doesn't this invalidate the conclusion drawn from Fig 4a, that is, the evaluation of ERA-I and ERA5 through GLEAM? See also comment (7) above.
**As replied to comment #7, the authors are convinced that forcing GLEAM with better inputs, can only lead to better validation statistics against in situ, when done over a large number of sites and for long-term periods. Hence, we think the statements are thus correct.**

**Changes in the manuscript:** no changes.

11. P11/L3-8: One aspect that may come into play here is that GLEAM+ERA5 is an off-line (land-only) modeling system that does not permit feedback whereas ERA5 is a coupled land-atmosphere modeling system. This may be related to a finding by Draper et al (2018): "Finally, the SH results for MERRA-Land are troubling. While MERRA-Land did have the desired reduction in the LH biases compared to MERRA (to 1 W/m2 in the global land annual average), it also had a compensating, and much larger, increase in the SH bias (up to 15W m22 in the global land average)" [beginning of p689

of Draper et al. 2018]. See also comment 5) above about the need for a better integration of the results of the present paper into the literature context.

**We fully agree with the referee that this is a main difference between both modelling approaches that may strongly affect the simulated fluxes. We also thank the reviewer for the suggested paper that fits well within the current manuscript.**

Changes in the manuscript: **this finding will be more discussed in detail and be put into better perspective within the existing literature.**

12. P11/L26-P12/ L2: There is no discussion of Fig 9! In this paragraph, insert explicit references to Figs 7, 8, and 9 in the relevant place within the paragraph. E.g., reference Fig 7 in P11/L28, reference Fig 8 in P11/L33. This reveals that Fig 9 has not been discussed.
**We thank the referee for picking this up and will extend the discussion of Figures 7–9.**

Changes in the manuscript: **the discussion on Figures 7–9 will be extended in the revised version of the manuscript.**

13. In some cases the results are overstated. E.g., P12/L22: "The improvements are less clear…" suggests that there are some (hard-to-see) improvements, when in fact the results are neutral at best. P14/L3-4: The statement here is not consistent with the results of Fig 4 that show that ERA-I estimates of the sensible heat flux and Bowen ratio are better than those of ERA5.
**The authors will have a detailed look at the conclusions again to soften some statements and to align the conclusions better with the results discussed in the remainder of the manuscript.**

Changes in the manuscript: **The manuscript will be screened for conclusions that might be too strong.**

14. The tower validation results should come with some measure of statistical significance or error bars. Are the improvements, that is, the small shifts in the distributions of the metrics as shown in the violin plots, meaningful?
**We agree with the reviewer that differences in quality are sometimes marginal, although often consistent. Although relying on assumptions on its own, we agree that a measure of statistical significance would be useful to report.**

Changes in the manuscript: **a measure of statistical significance will be included in the revised version of the manuscript.**

**Minor comments:**

1. P2/L13: Reichle et al. 2017 is a reference primarily for MERRA-2 land surface estimates. MERRA-2 which is a full atmospheric reanalysis, similar to ERA-I and ERA5. In this place, however, the authors are here referring to land-only reanalyses, such as ERA-Interim/Land and MERRA-Land. For the latter, Reichle et al. (2011) is a better reference. Note that there is *not* a land-only reanalysis associated with MERRA-2.
Reichle, R. H., et al. (2011), Assessment and enhancement of MERRA land surface hydrology estimates, Journal of Climate, 24, 6322-6338, doi:10.1175/JCLI-D-10-05033.1.
**We thank the referee for the suggested correction.**

Changes in the manuscript: the reference will be updated.

2.  P2/L26-30: The text here is about very geographically limited results (southern Antarctic peninsula) or sea-ice, which is not the focus of the present paper. I suggest deleting this text or moving it further down. It confuses the reader by distracting from the focus of the paper on the global land surface turbulent fluxes.
    **The main idea of this paragraph is to give a brief overview of studies that have already evaluated the quality of ERA5, irrespective of the scientific field, and to indicate that – although the number of studies is rather limited – results generally show a high quality of ERA5 compared to other datasets. We agree that these studies had a different focus than the current manuscript, but the authors believe that this short discussion is still relevant.**

    Changes in the manuscript: **no changes.**

3.  P4/L19: On first reading, I completely missed the term "standardised" here and later got confused about the lack of dimensions/units in the graphics. In many papers, anomalies from the seasonal cycle are examined without standardisation, i.e., they are dimensional anomalies. There is nothing wrong per se with the standardised anomalies, but please make it clearer that you are focusing on dimensionless anomalies.
    **Thanks for pointing this out.**

    Changes in the manuscript: **the use of standardised anomalies and the reason for this choice will be better described in the manuscript.**

4.  P10/L31: typo: "we should emphasis" –> "we should emphasise"
    **We thank the referee for their detailed look at the paper and picking this up.**

    Changes in the manuscript: **the typo will be corrected.**

5.  P12/L16+L20: The numbers referenced here contradict the numbers in the graphic.
    **We thank the referee for picking this up, the numbers were indeed incorrectly reported.**

    Changes in the manuscript: **the correct numbers will be reported in the revised version of the manuscript.**

6.  Caption Fig 2: Replace "For MD, the distribution of beta..." with "The distribution of the MD of beta..."???
    **We thank the referee for the suggestion.**

    Changes in the manuscript: **the captions will be updated per suggestion of the reviewer.**

7.  Caption Fig 7: "...between the absolute bias in ERA5 and ERA-I;"??? Maybe I'm misunderstanding this, but I think the bottom panel shows abs(bias(ERA-I)) minus abs(bias(ERA5), that is, the sign of the abs bias difference is different from what the caption suggests.
    **We thank the reviewer for pointing this out. The caption should indeed read: "The bottom map represents the difference between the absolute bias in ERA-I and ERA5; hence, green colors represent lower bias in ERA5 than in ERA-I."**

Changes in the manuscript: the caption will be corrected.

8. Caption Fig 10: "...versus ERA-I (squares)"??? Should this read "...versus ERA-I (triangles)"???
**We thank the referee again for his detailed look at the figures, the symbols were indeed wrongly referenced in the caption.**

Changes in the manuscript: the cation will be updated per suggestion of the reviewer.

9. Figure 11 needs units on the colorbars. I also suggest making the graphic bigger so it can be read more clearly in a hard copy for the next round of reviews.
**The authors agree that the figures were too small and that units need to be included on the colorbar.**

Changes in the manuscript: the size of the figure will be increased and units will be included.

---

## Author Comment (AC2) · 24 Apr 2020

RC2

**General comments:**
This is a very interesting and useful study of the representation of the land surface energy budget in European global atmospheric reanalyses. A large number of diverse in situ observations are used to benchmark several simulations at a global scale. Overall, the paper is well written, apart from the mixing of results and discussion/interpretation. Quality of some Figures could be improved. Colour scales are sometimes confusing as "green" tends to look blue. Violin plots are useful but do not provide a point by point comparison. Could all the corresponding scatter plots be given in a Supplement? A discussion on the impact of land cover is lacking. Recommendation: minor revisions.

**We would like to thank the referee for reviewing the paper and giving some interesting comments and feedback. Below, we give a point-to-point reply to the comments posted by the reviewer and list the changes that will be implemented in the manuscript.**

**Regarding the scatterplots, the authors believe that the violin plots together with the discussion of the results should give a sufficiently detailed understanding of the results and prefer not to include these addition figures.**

**Particular comments:**

1. P. 1, Title: should be more specific. For example: "Evaluating the land surface energy partitioning in European global atmospheric reanalyses".

   **We believe the reviewer means 'less specific'. The authors prefer to keep the title as is, as we want to emphasise that the focus of the paper is on the evaluation of the newest state-of-the art ERA5 reanalysis, rather than European reanalyses in general. Note that ERA-Interim only serves as a benchmark here to show the improvements. We believe mentioning the model/dataset in the title is in addition a requirement at GMD.**

   **Changes in manuscript:** **no changes**

2. P. 3, L. 1-2 ("perform better than ERA5"): any reference on this?

   **As this sentence builds upon the previous sentence, the reference supporting this statement is Urraca et al. (2018).**

   **Changes in manuscript:** **the reference will be cited again to make this clear.**

3. P. 4, L.7: I would be more specific. For example: "the more evolved HTESSEL land surface model in ERA5".

   **We agree with the referee that this sentence should be updated.**

   **Changes in manuscript:** **the sentence will be updated per suggestion of the reviewer.**

4. P. 4, L. 19: could you explain how these anomalies are defined and calculated?

   **The standardised anomalies are simply calculated by subtracting from the raw time series (1) the climatological expectation (i.e. the mean of the variable under consideration over at least 5 years for a certain time step) and (2) dividing by the standard deviation of that climatological expectation.**

Changes in manuscript: a brief description of this procedure will be included in the revised version of the manuscript.

5.  P. 5, L. 1-3: It seems that a key issue was not addressed. Land cover type in ERA5 may not correspond to the tower's one. E.g. a grassland Fluxnet site may be located in an ERA5 grid cell mainly covered by forests. How did you handle this?
    **We agree with the referee that this is an issue in the in situ evaluation strategy, as it is always the case in comparisons of grid cell values to in situ data. The mismatch in spatial footprint between the in situ measurement and the grid cell of the models typically leads to an overestimation of the actual error, often referred to as the representativeness error. In this study, we did not apply any filtering to maximise the representativeness of the in situ measurement – and hence to minimise this representativeness error – for the grid cell of the model. However, we do agree that this issue should be explicitly mentioned in the discussion of the results.**

    Changes in manuscript: **this issue will be highlighted in the discussion of the results.**

6.  P. 6, L. 5: could you define "non-overlapping moving windows"?
    **Non-overlapping (moving average) windows refer to the fact that the time windows used for calculating the averaged quantities do not intersect (e.g. Dehghani et al., 2019). Moving windows are commonly calculated for all data points of a time series, i.e. for a simple example with a window length of 5 months, the centered moving average (window) of March contains data from January, February, March, April and May, whereas the moving average centered on April is based on nearly the same data, except ranging from February to June (i.e., adjacent moving averages share some data, and are thus 'overlapping'). In this given example, the 'next' non-overlapping window would be centered on August (June–October), as the time window used for the centered average of March and this time window do not intersect.**

    Changes in manuscript: **this will be clarified in the text to make this more clear.**

7.  P. 6, L. 12 (G as a fixed fraction of Rn): Could this explain the poor scores obtained for sensible heat flux in Figure 8? The soil heat flux is related to soil properties and can be influenced by sensible heat exchange with rainwater (e.g. Zhang et al. [https://doi.org/10.5194/acp-19-5005-2019](https://doi.org/10.5194/acp-19-5005-2019)).
    **As described in the manuscript at page 12 lines 3–7, the results for the sensible heat flux should indeed be interpreted with care as they are (among others) affected by this specific assumption. However, the magnitude of the ground heat flux at daily scales is often substantially smaller than that of the other fluxes, so the authors expect only a minor impact of this assumption on the analyses.**
    **In addition, the approximation used in this study to calculate the ground heat flux is not that uncommon as the ground heat flux is typically strongly correlated to net radiation (see e.g. Kustas and Daughtry, 1990; Santanello and Friedl, 2003). Also note that, although we do not explicitly account for the effect of soil properties on the ground heat flux, we do account for the land cover type, as described in Miralles et al. (2011) and Martens et al. (2017) and the response to comment #8.**

    Changes in manuscript: **the calculation of the ground heat flux will be described in more detail in the revised version of the manuscript and references will be added.**

8. P. 6, L. 12 ("land cover"): which land cover? Is it the land cover used in the model?
The ground heat flux is calculated in this paper as described in Miralles et al. (2011) and Martens et al. (2017). In essence, the ground heat flux is calculated as a fixed fraction of net radiation, depending on the sub-pixel land cover heterogeneity. The latter is parameterised by the MOD44B Vegetation Continuous Fields product, describing each grid cell as a fraction of tall vegetation (e.g. forests), low vegetation (e.g. grasslands), and bare soil. For the fraction of tall vegetation, the ground heat flux is 10% of the net radiation, while for the fractions of low vegetation and bare soil the corresponding percentages are 20% and 35% (Miralles et al., 2011). In the end, the fraction of net radiation assumed to be converted into the ground heat flux is the weighted average of the former percentages with the fractional land covers.

Changes in manuscript: the calculation of the ground heat flux will be described in more detail in the revised version of the manuscript and references will be added.

9. P. 8, L. 3: Is there an impact of the land cover type?
The authors have tried to relate improvements/degradations from ERA-Interim to ERA5 to different ancillary data sets like land cover, elevation, and climate, but no conclusive results were obtained. Given the uncertainties in such an analysis, the authors have chosen not to further discuss these results.

Changes in manuscript: no changes

10. P. 8, L. 17: Seasonality removal should be described is chapter 2.
As replied to comment #4, the procedure will be described in more detail in the revised version of the paper.

Changes in manuscript: a brief description of this procedure will be included in the revised version of the manuscript.

11. P. 8, L. 28: what about Fluxnet site distribution in terms of vegetation types?
As replied to comment #8, the FLUXNET sites are indeed not well-distributed across vegetation types and climate.

Changes in manuscript: this issue will be further emphasised in the revised version of the paper.

**Editorial comments (Figures):**

1. Figure 1: Sites cannot be easily spotted. Colors of dots and background should be changed. What about land cover types? Format of the subfigure on the right should be consistent with format of Figure 3.
We agree with the reviewer that the details on the figure are hard to read.

Changes in manuscript: the figure will be updated to improve the readability.

2. Figure 6: green or blue?
We agree with the reviewer that the colours might be confusing for some readers, but we think this is comment is purely linguistic and that the figures are clear.

3. Figure 10 (top subfigures): meaning of the red lines? These metrics are a bit obscure. Why not comparing scatterplots of ABL heights?
**As described in the caption, the solid lines represent the median and inter-quartile range (green for ERA5 and red for ERA-Interim). We appreciate the suggestion. The reason why diurnal changes in temperature, humidity and ABL height are compared – as opposed to afternoon temperature, humidity and ABL height – is to reduce the influence of errors in the morning initial conditions. This in addition increases the comparability of the results reported by Wouters et al. (2019).**

Changes in manuscript: **a legend will be added to the figure.**

4. Figure 11: Not readable. Difference figures should be expanded. Green or blue?
**The authors agree that the figures were too small.**

Changes in manuscript: **the size of the figure will be increased so it covers the entire two columns of the manuscript.**

References:

1. Dehghani, A. et al.: A Quantitative Comparison of Overlapping and Non-Overlapping Sliding Windows for Human Activity Recognition Using Inertial Sensors. Sensors, 19, 5026, 2019.
2. Kustas, W. P. and Daughtry, C. S. T.: Estimation of the soil heat flux/net radiation ratio from spectral data, Agric. For. Meteorol., 49, 205–233, 1990.
3. Martens, B. et al.: GLEAM v3: Satellite-based land evaporation and root-zone soil moisture, Geosc. Model Dev., 10, 1903–1925, 2017.
4. Miralles, D.G. et al.: Global land-surface evaporation estimated from satellite-based observations, Hydrol. Earth Sys. Sc., 15, 453–369, 2011.
5. Santanello, J. A. and Friedl, M. A.: Diurnal Covariation in Soil Heat Flux and Net Radiation, J. Appl. Meteor., 42, 851–862, 2003.
6. Wouters H., et al.: Atmospheric boundary layer dynamics from balloon soundings worldwide: CLASS4GL v1.0, Geosc. Model Dev., 12, 2139–2153, 2019.

---

## Author Response (AR1)

**GMD-2019-315**

Dear GMD editor,

First, we would like to thank the associate editor and reviewers for handling our manuscript. Enclosed to this letter, a revised version of our manuscript GMD-2019-315 may be found. We believe we have addressed all comments raised by the two referees and modified the manuscript accordingly. The most important changes can be found in Sect. 2, where we have re-structured the entire content – per suggestion of reviewer 1 – to better separate the description of the data and methods used for evaluation. In addition, in Sect. 2, we now also introduce the method used for statistical testing, from which the results are further discussed in Sect. 3.

Finally, in addition to the corrections suggested by the two referees, some minor textual issues have been resolved in the revised version of the manuscript.

Below, a list of the comments per reviewer is given, together with our reply and a description of the changes in the manuscript in **bold fonts**. Please note that page and line numbers refer to the new version of the manuscript, unless explicitly mentioned. We hope that this revised version of the manuscript is eligible for publication in GMD. Attached to this letter, you may also find a marked-up version of the revised manuscript.

Brecht Martens,

On behalf of all co-authors.

**RC1 (published online: 11/03/2020):**

The paper investigates the skill of the land surface energy partitioning in the ERA5 reanalysis. For reference, the ERA5 skill is compared to that of its predecessor, ERA Interim (ERA-I). Skill is determined in several different ways: (1) directly vs flux tower measurements, (2) vs fluxes from water and energy balance estimates, (3) by driving the GLEAM land surface scheme with reanalysis data and validating the output vs. flux tower measurements, and, finally, (4) by driving the CLASS4GL boundary layer model with reanalysis data and validating the output vs. balloon observations. The authors find that ERA5 land surface energy partitioning is generally improved over that of ERA-I. In particular, the overestimation of the latent heat flux in ERA-I is reduced (but not eliminated) in ERA5.

The paper is of interest to GMD readers and makes an important contribution by documenting the quality of ERA5 land surface estimates. By and large, the writing and graphics are clear and concise, and the conclusions drawn from the study are supported by the results. I recommend eventual publication of the paper in GMD provided the authors address the comments below. It would be particularly helpful to include other reanalysis data in the comparison.

We thank the referee for reviewing the paper and providing us with useful comments, feedback, and corrections. Below, we give a point-to-point reply to the comments posted by the reviewer and describe the changes that have been implemented in the revised version of the manuscript. Please note that page and line numbers refer to the new version of the manuscript, unless explicitly mentioned.

Major comments (no particular order):

1. The title and introduction do not make it sufficiently clear that the turbulent fluxes investigated here are for the land only. I suggest changing the title to: "Evaluating the \*land\* surface energy partitioning in ERAS" and occasionally replacing the expression "surface [..] fluxes" with "land surface [..] fluxes", e.g., P3/L15, P4/L3, and probably a few more places.

The authors agree with the referee and have updated the title as suggested. Also, the term "land-surface fluxes" is now introduced more often into the text.

Changes in manuscript: The title of the manuscript has been changed to "Evaluating landsurface energy partitioning in ERAS".

2. P4/L1: Aren't there advances in land data assimilation from ERA-I to ERA5? And do these not matter for the quality of the land surface turbulent flux estimates? Land data assimilation in ERA-I and ERA5 and its impact on land surface estimates should be included briefly in the Introduction and further discussed in the Results and Discussion section.

The authors agree with the referee that changes in the land data assimilation system, and their potential influence, were not described in sufficient detail in the manuscript.

Changes in manuscript: In Sect. 2.1 (P4-L1-9), we now highlight that changes in the land data assimilation system might also have an important impact on the land-surface turbulent heat fluxes. In addition, in Sect. 3.1.1 (P11-L6-10), we also hypothesise that the more advanced land data assimilation system – together with the superior land-surface model implemented in the IFS – are behind the enhanced surface energy partitioning.

3. Section 2 is a somewhat odd mix of "methods" and "data". E.g., the FLUXNET 2015 section 2.2 includes discussion of how the climatology is derived for the computation of the standardised anomalies, but this would also apply to the other datasets (incl. ERA-I and ERA5). The entire section needs to be reorganized and be more clearly separated into "Data" and "Methods".
The authors agree that this would impress the reordability of the menuagrint.

The authors agree that this would improve the readability of the manuscript.

Changes in the manuscript: The authors have re-organised Sect. 2 per suggestion of the reviewer, aiming at a better separation of the methods and data.

The violin plots are great visual tools, but I assume their construction involves is some fitting of the distributions. These details should be in the "Methods" section.
 The method used to construct the violin plots involves a kernel density estimation approach where the band width is calculated according to Scott (1979).

Changes in the manuscript: The method for constructing the violin plots is now described in more detail in the revised version of the manuscript. We refer to Sect. 2.7.1 for more details; there we describe multiple aspects of the validation strategy.

 There are gaps in the literature discussion. E.g., Draper et al (2018) is a highly relevant assessment of reanalysis estimates of land surface energy flux estimates, incl ERA-I. Draper, C. S., R. H. Reichle, and R. D. Koster (2018), Assessment of MERRA-2 Land Surface Energy Flux Estimates, Journal of Climate, 31, 671-691, doi:10.1175/JCLI-D-17-0121.1.
 The authors thank the reviewer for this suggestion.

Changes in the manuscript: The paper by Draper et al. (2018) is now referenced in the discussion of the results. More in particular, the paper is cited in Sect. 3.2 where the biases in ERA5 are discussed and in Sect. 3.4, where the globally-averaged land-surface turbulent heat

fluxes are assessed.

6. Related to (5), the present paper only investigates ERA-I and ERA5. The paper would be considerably more relevant to readers if these ECMWF products were assessed along with at least one or two other, major reanalysis products (such as MERRA-2).

The authors thank the referee for this suggestion and fully agree that this would be an interesting analysis. However, the focus of the paper is to make a first assessment of the quality of the surface turbulent fluxes in ERA5 against different observation-based data sets. Including other reanalyses – next to ERA-I, which is there just to show improvements of ERA5 upon its predecessor – would deviate the focus from ERA5, and turn this manuscript in the direction of a comparison paper. We agree on the value of such a study in the future; yet, as we say, such a comparison feels beyond the scope of this work.

**Changes in the manuscript: No changes have been implemented in the manuscript.**

7. If I understood this correctly, the GLEAM and CLASS4GL analyses work as follows: (i) use ERA-I and, separately, ERA5 to force GLEAM (or CLASS4GL), then (ii) evaluate the results against tower (or balloon) measurements. This approach depends on the assumption that GLEAM and CLASS4GL are very good models, or at least that they do not have compensating errors. If, say, errors in GLEAM were to compensate for errors in ERA-I, forcing GLEAM with a better reanalysis isn't necessarily going to

deliver better GLEAM outputs. Similarly for CLASS4GL. This is a major caveat that needs to be discussed prominently in the paper.

The authors agree that this type of analysis comes with assumptions, yet it still provides useful insights in the quality of ERA5. Unless structural errors in GLEAM and CLASS4GL are somehow (anti-)correlated, forcing either of the models with better inputs can only lead to better model output. This would therefore yield better validation results when compared against independent in situ observations, as long as it is done systematically at a large number of sites. Given the strong differences between the model concepts and assumptions of GLEAM and CLASS4GL, the authors believe that both model frameworks are sufficiently independent (i.e. their errors are sufficiently uncorrelated) to trust that a higher accuracy in the forcing data should reflect on better match to the in situ measurements after this modelling chain.

Changes in the manuscript: This caveat in the analysis is now briefly mentioned in Sect. 3.1.2 (P12-L4-11).

8. Figures A.1-A.4 are oddly placed. Either they need to be put into a proper Appendix, with discussion in the appendix as well, or they need to placed in a separate "Supplementary Information" document. As assembled, Figures A.1-A.4 are simply out of order.

According to the guidelines of Copernicus, the authors think that Figures A1-A4 should actually be included into the manuscript as appendices:

"Additional figures, tables, as well as technical and theoretical developments which are not critical to support the conclusion of the paper, but which provide extra detail and/or support useful for experts in the field and whose inclusion in the main text would disrupt the flow of descriptions or demonstrations may be presented as appendices. These should be labelled with capital letters: Appendix A, Appendix B etc. Equations, figures and tables should be numbered as (A1), Fig. B5 or Table C6, respectively. Please keep in mind that appendices are part of the manuscript whereas supplements (see below) are published along with the manuscript."

"Supplementary material is reserved for items that cannot reasonably be included in the main text or as appendices. These may include short videos, very large images, maps, CIF files, as well as short computer codes such as matlab or python script."

The text above has been exactly cited as it appears at https://www.geoscientific-model-development.net/for\_authors/manuscript\_preparation.html (last visit June, 1st 2020).

**Changes in the manuscript: No changes have been implemented in the manuscript, but we will follow the editorial advice on this.**

9. Figures 2 and 4 are a confusing mix of dimensional metrics (MD for raw fluxes) and unit-less metrics (MD for raw Bowen ratio, and MAD and R for standardised anomalies). These figures also lack basic information about a dimensional 2nd-order metric such as MAD or RMSE for dimensional (raw or anomaly) variables. The readers are going to want to know typical MAD or RMSE values for fluxes in units of W/m2, incl. and/or excl. the seasonal cycle. I suggest revisiting the assembly of Figures 2, 4, A.1-A.4. The paper would be much easier to follow and more informative if, say, one figure includes only (dimensional) metrics computed from raw time series and another figure includes only (non-dimensional) metrics from standardised anomaly time series.

The primary reason to focus on the evaluation of anomalies is to minimise the effect of the strong seasonal cycle in the turbulent fluxes, which might mask important differences between the quality of ERA5 and ERA-I. Then, we prefer to focus on standardised anomalies to allow direct comparison of metrics from the different turbulent fluxes that typically range in a different order of magnitude. Needless to say, calculating the Mean Difference on anomaly time series is not needed, as anomalies are mean zero; hence we report the Mean Difference calculated on raw time series.

Nonetheless, we do agree with the referee that readers might be interested in the statistics calculated on raw time series as well. Note that, therefore, we report these corresponding statistics in Figures A1 and A4.

Finally, the authors believe that the content of the figures is described with sufficient detail in the captions and corresponding text to avoid confusion.

Changes in the manuscript: We better motivate our choice to evaluate standardised anomalies in Sect. 2.7.1 (P8-L26-P9-L2) of the revised version of the manuscript.

10. P10/L15-16: "Figure 4b shows that ERA5 is better at capturing..." Doesn't this invalidate the conclusion drawn from Fig 4a, that is, the evaluation of ERA-I and ERA5 through GLEAM? See also comment (7) above.

As replied to comment #7, forcing GLEAM with better inputs can only lead to better validation statistics against in situ, when done over a large number of sites and for long-term periods. This is especially true when focussing on inferences such as anomaly correlations, and could only be rebutted when referring to e.g., long-term biases. Hence, we think the statements are correct.

Changes in the manuscript: No changes have been implemented in the manuscript.

11. P11/L3-8: One aspect that may come into play here is that GLEAM+ERA5 is an off-line (land-only) modeling system that does not permit feedback whereas ERA5 is a coupled land-atmosphere modeling system. This may be related to a finding by Draper et al (2018): "Finally, the SH results for MERRA-Land are troubling. While MERRA-Land did have the desired reduction in the LH biases compared to MERRA (to 1 W/m2 in the global land annual average), it also had a compensating, and much larger, increase in the SH bias (up to 15W m22 in the global land average)" [beginning of p689 of Draper et al. 2018]. See also comment 5] above about the need for a better integration of the results of the present paper into the literature context.

We fully agree with the referee that this is an important difference between both modelling approaches that may strongly affect the simulated fluxes. We also thank the reviewer for the suggested paper that fits well within the current manuscript.

Changes in the manuscript: The fact that ERA5 is a fully-coupled system, while GLEAM is an offline land-surface model is now highlighted in the manuscript (P13-L29-31) and reference to Draper et al. (2018) is included in several contexts (Sect. 3.2 and 3.4).

 P11/L26-P12/ L2: There is no discussion of Fig 9! In this paragraph, insert explicit references to Figs 7, 8, and 9 in the relevant place within the paragraph. E.g., reference Fig 7 in P11/L28, reference Fig 8 in P11/L33. This reveals that Fig 9 has not been discussed.
 We thank the referee for picking this up.

Changes in the manuscript: Given that the conclusions for the Bowen ratio largely follow from the discussion on the turbulent fluxes, we briefly refer to Fig. 9 now in P15-L5-7.

In some cases the results are overstated. E.g., P12/L22: "The improvements are less clear..." suggests that there are some (hard-to-see) improvements, when in fact the results are neutral at best. P14/L3-4: The statement here is not consistent with the results of Fig 4 that show that ERA-I estimates of the sensible heat flux and Bowen ratio are better than those of ERA5.

The authors will have a detailed look at the conclusions again to soften some statements and to align the conclusions better with the results discussed in the remainder of the manuscript.

Changes in the manuscript: The manuscript will be screened for conclusions that might be too strong.

14. The tower validation results should come with some measure of statistical significance or error bars. Are the improvements, that is, the small shifts in the distributions of the metrics as shown in the violin plots, meaningful?

We agree with the reviewer that differences in quality are sometimes marginal, although often consistent. Although relying on assumptions of its own, we agree that reporting a measure of statistical significance would be useful. Therefore, we now test the differences in the anomaly correlations and the Mean Difference (MD) of the main experiments (i.e. ERA5 vs ERA-I and GLEAM vs ERA) for statistical significance (at the 5% significance level). Note that no tests were introduced for the Mean Absolute Difference (MAD), as no analytical solutions to calculate the confidence intervals are available for this metric. However, we believe that the conclusions drawn from the MAD largely follow the ones from the MD and Pearson Correlation.

Changes in the manuscript: A detailed description on the method used for statistical testing is given in Sect. 2.7.1 (P9-L11-19). Results of the statistical tests are included throughout the discussions in Sect. 3.1.1 and 3.1.2.

**Minor comments:**

 P2/L13: Reichle et al. 2017 is a reference primarily for MERRA-2 land surface estimates. MERRA-2 which is a full atmospheric reanalysis, similar to ERA-I and ERA5. In this place, however, the authors are here referring to land-only reanalyses, such as ERA-Interim/Land and MERRA-Land. For the latter, Reichle et al. (2011) is a better reference. Note that there is \*not\* a land-only reanalysis associated with MERRA-2.

Reichle, R. H., et al. (2011), Assessment and enhancement of MERRA land surface hydrology estimates, Journal of Climate, 24, 6322-6338, doi:10.1175/JCLI-D-10-05033.1. **We thank the referee for the suggested correction**.

Changes in the manuscript: The reference has been updated.

 P2/L26-30: The text here is about very geographically limited results (southern Antarctic peninsula) or sea-ice, which is not the focus of the present paper. I suggest deleting this text or moving it further down. It confuses the reader by distracting from the focus of the paper on the global land surface turbulent fluxes.

The main idea of this paragraph is to give a brief overview of studies that have already evaluated the quality of ERA5, irrespective of the scientific field, and to indicate that – although the number of studies is rather limited – results generally show a high quality of ERA5 compared to other datasets. We agree that these studies had a different focus than the current manuscript, but the authors believe that this short discussion is still relevant.

Changes in the manuscript: No changes have been implemented in the manuscript.

3. P4/L19: On first reading, I completely missed the term "standardised" here and later got confused about the lack of dimensions/units in the graphics. In many papers, anomalies from the seasonal cycle are examined without standardisation, i.e., they are dimensional anomalies. There is nothing wrong per se with the standardised anomalies, but please make it clearer that you are focusing on dimensionless anomalies.

Thanks for pointing this out.

Changes in the manuscript: The use of standardised anomalies and the reason for this choice are now highlighted in Section 2.7.1 of the revised manuscript.

4. P10/L31: typo: "we should emphasis" -> "we should emphasise"
 We thank the referee for their detailed look at the paper and picking this up.

Changes in the manuscript: The typo has been corrected.

P12/L16+L20: The numbers referenced here contradict the numbers in the graphic.
 We thank the referee for picking this up, the numbers were indeed incorrectly reported.

Changes in the manuscript: This has been corrected.

 Caption Fig 2: Replace "For MD, the distribution of beta..." with "The distribution of the MD of beta..."???
 We thank the referee for the suggestion

We thank the referee for the suggestion.

Changes in the manuscript: The caption has been updated per suggestion of the reviewer.

Caption Fig 7: "...between the absolute bias in ERA5 and ERA-I;"??? Maybe I'm misunderstanding this, but I think the bottom panel shows abs(bias(ERA-I)) minus abs(bias(ERA5), that is, the sign of the abs bias difference is different from what the caption suggests.
 We thank the reviewer for pointing this out. The caption should indeed read: "The bottom map represents the difference between the absolute bias in ERA-I and ERA5; hence, green colors represent lower bias in ERA5 than in ERA-I."

Changes in the manuscript: The caption has been corrected.

8. Caption Fig 10: "...versus ERA-I (squares)"??? Should this read "...versus ERA-I (triangles)"???

We thank the referee again for his detailed look at the figures, the symbols were indeed wrongly referenced in the caption.

Changes in the manuscript: The caption has been updated per suggestion of the reviewer.

 Figure 11 needs units on the colorbars. I also suggest making the graphic bigger so it can be read more clearly in a hard copy for the next round of reviews.
 The authors agree that the figures were too small and that units need to be included on the colorbar.

Changes in the manuscript: The size of the figure has been increased and units have been added.

**RC2 (published online: 24/03/2020):**

**General comments:**

This is a very interesting and useful study of the representation of the land surface energy budget in European global atmospheric reanalyses. A large number of diverse in situ observations are used to benchmark several simulations at a global scale. Overall, the paper is well written, apart from the mixing of results and discussion/interpretation. Quality of some Figures could be improved. Colour scales are sometimes confusing as "green" tends to look blue. Violin plots are useful but do not provide a point by point comparison. Could all the corresponding scatter plots be given in a Supplement? A discussion on the impact of land cover is lacking. Recommendation: minor revisions.

We would like to thank the referee for reviewing the paper and giving some interesting comments and feedback. Below, we give a point-to-point reply to the comments posted by the reviewer and list the changes that will be implemented in the manuscript.

Regarding the scatterplots, the authors believe that the violin plots together with the discussion of the results should give a sufficiently detailed understanding of the results and prefer not to include these additional figures.

Particular comments:

1. P. 1, Title: should be more specific. For example: "Evaluating the land surface energy partitioning in European global atmospheric reanalyses".

We believe the reviewer means 'less specific', maybe. The authors prefer to emphasise that the focus of the paper is on the evaluation of the state-of-the art ERA5 reanalysis, rather than European reanalyses in general. Note that ERA-Interim only serves as a benchmark here to show the improvements. In addition, we believe mentioning the model/dataset in the title a requirement at GMD. However, note, we did slightly modify the title to highlight that the manuscripts focusses on land-surface fluxes only.

Changes in manuscript: The title of the manuscript has been changed to "Evaluating landsurface energy partitioning in ERAS".

P. 3, L. 1-2 ("perform better than ERA5"): any reference on this?
 As this sentence builds upon the previous sentence, the reference supporting this statement is Urraca et al. (2018).

Changes in manuscript: The reference has been cited again to support this statement (P2-L30-35).

3. P. 4, L.7: I would be more specific. For example: "the more evolved HTESSEL land surface model in ERAS".

We agree with the suggestion of the referee.

Changes in manuscript: The sentence has been updated per suggestion of the referee.

 P. 4, L. 19: could you explain how these anomalies are defined and calculated? The standardised anomalies are simply calculated by subtracting from the raw time series (1) the climatological expectation (i.e. the mean of the variable under consideration over at least 5 years for a certain time step) and (2) dividing by the standard deviation of that climatological expectation.

Changes in manuscript: The calculation of the standardised anomalies is now described in Sect. 2.7.1 of the revised version of the manuscript.

5. P. 5, L. 1-3: It seems that a key issue was not addressed. Land cover type in ERA5 may not correspond to the tower's one. E.g. a grassland Fluxnet site may be located in an ERA5 grid cell mainly covered by forests. How did you handle this?

We agree with the referee that this is an issue in the in situ evaluation strategy, as it is always the case in comparisons of grid cell values to in situ data. The mismatch in spatial footprint between the in situ measurement and the grid cell of the models typically leads to an overestimation of the model error, and is often referred to as the representativeness error. In this study, we did not apply any filtering to maximise the representativeness of the in situ measurements – and hence to minimise this representativeness error. However, we do agree that this issue should be explicitly mentioned in the discussion of the results.

Changes in manuscript: This issue is explicitly highlighted in the revised version of the manuscript now (P8-L24-25, P12-L1-2, and P13-L23-25).

6. P. 6, L. 5: could you define "non-overlapping moving windows"?

Non-overlapping (moving average) windows refer to the fact that the time windows used for calculating the averaged quantities do not intersect (e.g. Dehghani et al., 2019). Moving windows are commonly calculated for all data points of a time series, i.e. for a simple example with a window length of 5 months, the centered moving average (window) of March contains data from January, February, March, April and May, whereas the moving average centered in April is based on nearly the same data, except ranging from February to June (i.e., adjacent moving averages share some data, and are thus 'overlapping'). In this given example, the 'next' non-overlapping window would be centered in August (June–October), as the time window used for the centered average of March and this time window do not intersect.

Changes in manuscript: This specific processing step is now better described in Sect. 2.3.1 (P6-L6-11).

P. 6, L. 12 (G as a fixed fraction of Rn): Could this explain the poor scores obtained for sensible heat flux in Figure 8? The soil heat flux is related to soil properties and can be influenced by sensible heat exchange with rainwater (e.g. Zhang et al. https://doi.org/10.5194/acp-19-5005-2019). As was described in the original manuscript at P12-L3-7, the results for the sensible heat flux should indeed be interpreted with care as they are (among others) affected by this specific assumption. However, the magnitude of the ground heat flux at daily scales is often substantially smaller than that of the other fluxes, so the authors expect only a minor impact of this assumption on the analyses.

In addition, the approximation used in this study to calculate the ground heat flux is not uncommon, as the magnitude of the ground heat flux typically scales with net radiation (see e.g. Kustas and Daughtry, 1990; Santanello and Friedl, 2003). Also note that, although we do not explicitly account for the effect of soil properties on the ground heat flux, we do account for the land cover type, as described by Miralles et al. (2011) and Martens et al. (2017) – see also the response to the following comment.

Changes in manuscript: The calculation of the ground heat flux is now described in detail in the Sect. 2.3.3 of the revised manuscript (P6-L17-26).

8. P. 6, L. 12 ("land cover"): which land cover? Is it the land cover used in the model?

The ground heat flux is calculated in this paper as described by Miralles et al. (2011) and Martens et al. (2017). In essence, the ground heat flux is calculated as a fixed fraction of net radiation, depending on the sub-pixel land cover heterogeneity. The latter is parameterised using the MOD44B Vegetation Continuous Fields product, describing each grid cell as a fraction of tall vegetation (e.g. trees), low vegetation (e.g. grass), and bare soil. For the fraction of tall vegetation, the ground heat flux is 10% of the net radiation, while for the fractions of low vegetation and bare soil the corresponding percentages are 20% and 35% (Miralles et al., 2011). In the end, the fraction of net radiation assumed to be converted into the ground heat flux at a given pixel is the weighted average of the former percentages considering the fractional land covers (MOD44B).

Changes in manuscript: The calculation of the ground heat flux is now described in detail in the Sect. 2.3.3 of the revised manuscript (P6-L17-26).

9. P. 8, L. 3: Is there an impact of the land cover type? The authors have tried to relate improvements/degradations from ERA-Interim to ERA5 to different ancillary data sets like land cover, elevation, and climate, but no conclusive results were obtained. Given the uncertainties in such an analysis, the authors have chosen not to further discuss these results.

Changes in manuscript: No changes have been implemented in the manuscript.

P. 8, L. 17: Seasonality removal should be described is chapter 2.
 As replied to comment #4, the procedure has been described in more detail in the revised version of the paper.

Changes in manuscript: The calculation of the standardised anomalies is now described in Sect. 2.7.1 of the revised manuscript.

P. 8, L. 28: what about Fluxnet site distribution in terms of vegetation types?
 As replied to comment #8, the FLUXNET sites are indeed not well-distributed across vegetation types and climate.

Changes in manuscript: The fact that FLUXNET sites are not uniformly-distributed across the global land surface is now mentioned several times in the manuscript (e.g. P4-L26-29, P11-L1-4, P18-L6-9).

**Editorial comments (Figures):**

 Figure 1: Sites cannot be easily spotted. Colors of dots and background should be changed. What about land cover types? Format of the subfigure on the right should be consistent with format of Figure 3.

We agree with the reviewer that the details on the figure are hard to read.

Changes in manuscript: The figure has been updated to increase the readability, by including zoomed-in panels on regions of interest.

2. Figure 6: green or blue?

We agree with the reviewer that the colours might be confusing for some readers, but we think this is comment is purely linguistic and that the figures are clear.

Changes in manuscript: No changes have been implemented in the manuscript.

3. Figure 10 (top subfigures): meaning of the red lines? These metrics are a bit obscure. Why not comparing scatterplots of ABL heights?

We appreciate the suggestion. As described in the caption, the solid lines represent the median and inter-quartile range (green for ERA5 and red for ERA-Interim). The reason why diurnal changes in temperature, humidity and ABL height are compared – as opposed to afternoon temperature, humidity and ABL height – is to reduce the influence of errors in the morning initial conditions. This in addition increases the comparability of the results, as discussed by Wouters et al. (2019).

Changes in manuscript: A legend has been added to the figure.

4. Figure 11: Not readable. Difference figures should be expanded. Green or blue? The authors agree that the figures were too small.

Changes in manuscript: The size of the figure has been increased and units have been added.

- Despite the importance of an accurate representation of the processes involved in the surface energy partitioning, at present and to the authors best knowledge, no studies have study has directly evaluated the partitioning of energy in ERA5 into the two major surface turbulent fluxes over land (i.e. the surface sensible and surface latent heat fluxlatent heat fluxes). As surface energy partitioning acts as a nexus between the land surface and atmosphere, such an analysis might provide useful insights to further improve the modelling of this coupled system, and to advance the quality of future reanalyses. Therefore, the
- 25 objective of this study is to evaluate the surface turbulent fluxes (and their ratio; i.e. the Bowen ratio) from ERA5 for the period 1983-2014-1983-2018 at different spatio-temporal resolutions. Several experiments are conducted using various observational data sets and modelling tools to evaluate the spatial and temporal variability of these variables the turbulent fluxes at different scales, ranging from point to catchment-scale and sub-daily to yearly scales. The paper is organised as follows: in Sect. 2 we describe the experimental set-up and the data sets used in this study, and provide a brief overview of the key differences
- 30 between ERA-I and ERA5. In Sect. 3 we describe the results of our experiments and discuss the quality of surface energy partitioning in both reanalyses; concluding remarks are summarised in Sect. 4.

**2 Methods Data and datamethods**

**2.1 ERA5 and ERA-I Reanalyses data**

ERA5 is the latest state-of-the-art reanalysis produced at ECMWF (Hersbach et al., 2020, 2018)(Hersbach et al., 2020), replacing the widely-used ERA-Ireanalysis (Dee et al., 2011). A first segment of the data set, covering the period 2010–2016,
was released early 2017, about a decade after the successful release of ERA-I. Compared to ERA-I, which uses IFS cycle 31r1, ERA5 is produced using an improved version of ECMWF's modelling and data assimilation system (IFS cycle 41r2) and ingests information from a substantially larger volume of improved observations, resulting in a high-quality reanalysis of global atmospheric, oceanic, and land-surface fields at hourly time steps, 137 vertical pressure levels, and at horizontal spatial horizontal resolution of approximately 31 km. Several of the changes relative to advancements upon ERA-I are ex-

[revised manuscript text omitted]

- and some key meteorological drivers of the turbulent fluxes - in ERA5 and ERA-I at both 3 hourly and daily temporal resolution for the period 1991–2014.

**2.3 Catchment water and energy-balance data**

Whenever If changes in water storage can be are neglected, the catchment-scale latent heat flux can be calculated as precipita-

5 tion minus river discharge; both averaged over a sufficiently long time period (Miralles et al., 2016; Liu et al., 2014; Wang and Dickinson, 2012; Miralles et al., 2011; Vinukollu et al., 2011)and by...By taking into account the latent heat of vaporisation and the density of water:

$$\lambda \rho E = \lambda \rho (P - Q),\tag{1}$$

where  $\lambda$  is the latent heat of vaporisation of water (assumed to be constant; 2260·103 J kg-1),  $\rho$  is the density of liquid water

[revised manuscript text omitted]

In summary, the evaporative fraction from both ERA-I and ERA5 is used to optimise the root-zone soil moisture in CLASS4GL in an iterative procedure as described in Wouters et al. (2019). The resulting output of the modelling framework is evaluated against the measurements As the temporal variability of the turbulent fluxes is strongly influenced by the seasonal

- 10 cycle of its main drivers at the scales considered in this experiment, the performance of the land-surface schemes in response to anomalous weather conditions (i.e. with respect to the seasonal cycle) might be masked when raw time series are analysed. As such, the evaluation of the turbulent fluxes against the FLUXNET data set will be done based on standardised anomalies to better evaluate the skill of the reanalyses in capturing the effect of specific meteorological conditions on the surface energy partitioning. Therefore, standardised anomalies of the turbulent fluxes are calculated (and Bowen ratio) from (1) the reanalyses.
- 15 (2) the GLEAM experiments, and (3) the eddy-covariance measurements prior to calculating validation metrics. Note that the calculation of standardised anomalies allows to directly compare the quality of the turbulent fluxes and the Bowen ratio, despite their different orders of magnitude.

Anomaly time series are calculated by (1) subtracting for each time interval the expected value (i.e. the climatology), calculated as the multi-annual average for that time interval, and (2) dividing by the standard deviation of the expectation.

20 To calculate climatologies of the eddy-covariance data, only FLUXNET sites with a minimum record length of five years are considered, resulting in 77 eddy-covariance towers for the evaluation of the anomaly time series (Fig. 1).

Using the standardised anomalies of the in situ eddy-covariance measurements as a reference, the Pearson correlation coefficient (R) and Mean Absolute Difference (MAD) of the reanalyses data sets and the estimates from GLEAM are calculated to evaluate their quality (Sect. 3.1.1). In addition, the Mean Difference (MD) of the raw data series is calculated to assess the

- 25 bias in the estimates. Metrics are visualised in violin plots constructed using a kernel density estimation approach with a band width calculated according to Scott (1979). For the MD and *R*, a 95% confidence interval is calculated at each FLUXNET site following the procedure outlined in De Lannoy and Reichle (2016). First, the temporal auto-correlation in both the reference and estimated time series is calculated to correct the degrees of freedom (Gruber et al., 2020). Second, a confidence interval is calculated at each FLXNET site assuming a normal distribution for *R* (after applying a Fisher Z-transformation to the time
- 30 series) and a Student *t*-distribution for the MD. Metrics are then assumed to be statistically different at the 5% significance level if their confidence intervals do not overlap. Note that we do not calculate confidence intervals for the MAD, as there are no analytical solutions available for this metric and the calculation thus requires a non-parametric approach relying on computationally heavy Monte Carlo simulations (Gruber et al., 2020). Finally, the confidence intervals for the MD and *R* are averaged across the FLUXNET data set and the average confidence interval is reported.

In a similar manner as for the GLEAM experiment, the simulations of CLASS4GL (Sect. 2.6) are validated against afternoon profiles from balloon soundings thereby providing an indirect and independent sourced from the IGRA data set (Sect. 2.4). However, the skill of CLASS4GL is evaluated based on the Root Mean Squared Error (RMSE) – rather than MAD – R, and MD, all calculated on raw time series, and results are visualised in Taylor plots.

**5 2.7.2 Evaluation using catchment energy-balance data**

Next to the evaluation of the surface energy partitioning in both reanalyses. Alltogether, the experiment investigates whether the partitioning provided by turbulent fluxes from ERA5 is more consistent with observed atmospheric boundary layer parameters than that provided by ERA-Iagainst 
[revised manuscript text omitted]
 improves the skill of CLASS4GL in simulating the diurnal variations in the atmospheric boundary layer; hence beneficial for boundary layer climate studies can lead to improved skill in

5 the diurnal ABL simulations by mixed-layer models such as CLASS.

Skill comparison of the CLASS4GL model (http://class4gl.eu; Wouters et al., 2019) for reproducing diurnal changes in boundary layer properties forced with surface evaporative fraction from ERA5 versus ERA-I. Shown are the tendencies of the mixed-layer height (dh/dt), potential temperature ( $d\theta/dt$ ) and specific humidity (dq/dt), which are assessed by comparison of model simulations against the IGRA sounding data between 1981 and 2015. The first (ERA5 forced) and second (ERA-I

10 forced) row show modeled versus observed data points (gray) and the corresponding median (green) and interquartile range (red) of the model. The 1–1 line is shown as a black dashed line. The last row indicates the model skill 
[revised manuscript text omitted]

|                      |             | $\frac{\lambda \rho E \text{ (3h)}}{W \text{ m}^{-2}}$ | $\frac{\lambda \rho E \text{ (24h)}}{W \text{ m}^{-2}}$ | $\frac{H(3h)}{W m^{-2}}$ | $\frac{H(24h)}{W m^{-2}}$ | β (24h)
≂  |
|----------------------|-------------|--------------------------------------------------------|---------------------------------------------------------|--------------------------|---------------------------|----------------------|
| MD                   | ERA5        | 9.27 (±0.080)                                          | 8.49 (±0.178)                                           | -2.60 (±0.010)           | -2.99 (±0.140)            | -0.56 (±0.013)       |
|                      | ERA-I       | 11.12 (±0.079)                                  | 10.29 (±0.180)                                          | -3.38 (±0.099)           | - 3.66 (±0.147)    | -0.69 (±0.012)       |
|                      | GLEAM+ERA5  | n.a.                                                   | -3.27 (±0.176)                                          | n.a.                     | - 5.83 (±0.153)    | -0.25 (±0.014)       |
|                      | GLEAM+ERA-I | n.a.                                                   | -3.76 (±0.179)                                          | n.a.                     | -10.14 (±0.158)           | -0.39 (±0.014)       |
| $\stackrel{R}{\sim}$ | ERA5        | $0.34 (\pm 0.002)$                                     | $\underbrace{0.41}_{(\pm 0.005)}$                       | $0.46(\pm 0.002)$        | 0.50 (±0.004)             | 0.39 (±0.006) |
|                      | ERA-I       | $\underbrace{0.31}_{(\pm 0.002)}$                      | $\underbrace{0.39(\pm 0.005)}$                          | $0.42 (\pm 0.002)$       | 0.45 (±0.004)             | 0.36 (±0.006)        |
|                      | GLEAM+ERA5  | n.a.                                                   | $0.35~(\pm 0.005)$                                      | n.a.                     | 0.45 (±0.005)             | 0.39 (±0.006) |
|                      | GLEAM+ERA-I | n.a.                                                   | 0.32 (±0.005)                                           | n.a.                     | 0.46 (±0.005)             | 0.40 (±0.007) |

Appendix A: Supplementary figures